# Adenosine 2A receptor and TIM3 suppress cytolytic killing of tumor cells via cytoskeletal polarization

Grace L. Edmunds[1], Carissa C. W. Wong[1], Rachel Ambler[1], Emily J. Milodowski[2], Hanin Alamir[1], Stephen J. Cross [3], Gabriella Galea [1], Christoph Wülfing [1,4 ✉] & David J. Morgan [1,4 ✉]

Tumors generate an immune-suppressive environment that prevents effective killing of tumor cells by CD8[+] cytotoxic T cells (CTL). It remains largely unclear upon which cell type and at which stage of the anti-tumor response mediators of suppression act. We have combined an in vivo tumor model with a matching in vitro reconstruction of the tumor microenvironment based on tumor spheroids to identify suppressors of anti-tumor immunity that directly act on interaction between CTL and tumor cells and to determine mechanisms of action. An adenosine 2A receptor antagonist, as enhanced by blockade of TIM3, slowed tumor growth in vivo. Engagement of the adenosine 2A receptor and TIM3 reduced tumor cell killing in spheroids, impaired CTL cytoskeletal polarization ex vivo and in vitro and inhibited CTL infiltration into tumors and spheroids. With this role in CTL killing, blocking $A_{2A}R$ and TIM3 may complement therapies that enhance T cell priming, e.g. anti-PD-1 and anti-CTLA-4.

[1] School of Cellular and Molecular Medicine, University of Bristol, Bristol BS8 1TD, UK. [2] Bristol Veterinary School, University of Bristol, Bristol BS40 5DU, UK. [3] Wolfson BioImaging Facility, University of Bristol, Bristol BS8 1TD, UK. [4] These authors jointly supervised this work: Christoph Wülfing and David J. Morgan. ✉email: Christoph.Wuelfing@bristol.ac.uk; D.J.Morgan@bristol.ac.uk

CD8+ cytotoxic T cells (CTL) have the ability to directly kill tumor target cells. Such killing requires effective priming of tumor antigen specific CD8+ T cells within the draining lymph nodes; differentiation into CTL; effective tumor infiltration and execution of the cytolytic effector function within the tumor microenvironment. However, solid tumors often generate an immune-suppressive environment with multiple often redundant immune-suppressive elements that prevent effective tumor cell killing. For the development of widely applicable curative cancer immunotherapies, multiple reagents with defined mechanisms of action that can be flexibly combined are required, at least some of which need to restore CTL killing within the tumor[1]. Here we have characterized two such reagents.

Inhibitory receptors, in particular CTLA-4, PD-1, TIGIT, TIM3, and LAG3, impair the anti-tumor immune response[2,3]. Blocking CTLA-4 and PD-1 is a cornerstone of current immunotherapy and has yielded great therapeutic success in many cancer types[4]. However, efficacy is limited to a subset of patients and few tumor types, and autoimmune side effects can be substantial. Mechanisms of action of PD-1 blockade are still being debated. Initially, it was widely assumed that blocking PD-1 would reactive tumor-infiltrating CTL (TIL) that had acquired a suppressed state characterized by enhanced PD-1 expression. However, PD-1 and also CTLA-4 are expressed not only on CTL but also on other T cell subtypes, other immune cell types, and even on tumor cells. Deletion of PD-1 in myeloid cells can enhance anti-tumor immunity more effectively than deletion in T cells[5]. In basal or squamous cell carcinoma patients, PD-1 blockade does not reactivate tumor-resident CTL but leads to infiltration of new CTL clones[6]. We have shown that treating mice with anti-PD-1 enhances anti-tumor immunity but treating TIL directly ex vivo does not enhance their function[7], further arguing for an effect independent of the direct CTL tumor cell interaction. Ligands of CTLA-4 are expressed on antigen-presenting cells in T cell priming but not commonly on tumor cells. Therefore, CTLA-4 blockade is more likely to enhance T cell priming rather than reactivate TIL. Potential depletion of regulatory T cells mediated by the anti-CTLA-4 Fc region, even though controversial, can also be expected to affect priming[8-11] and illustrates the wider functional consequences of Fc receptor engagement by antibodies against inhibitory receptors[12]. The effects of PD-1 and CTLA-4 blockade on T cell priming may be critical in the development of autoimmune side effects by allowing self-reactive T cells to activate. PD-1 and CTLA-4 blockade are most effective in patients that already have tumors with a substantial immune infiltrate rich in CD8+ T cells[13], limiting the applicability of PD-1 and CTLA-4 blockade across many cancer types. Thus, means to enhance anti-tumor immunity that focuses more strongly on (re)activation of CTL and/or enhancing tumor infiltration rather than on the priming of new T cell clones are of substantial interest in the development of a diverse combinatorial tool kit for curative cancer immunotherapy.

Key soluble mediators of tumor-mediated immune suppression are adenosine and prostaglandin E2 (PGE$_2$). Both use an increase in intracellular cyclic AMP (cAMP) as a key signaling mechanism and may, therefore, have overlapping functions[14,15]. PGE$_2$ strongly regulates dendritic cell biology and thus T cell priming[16]. Adenosine is generated by hydrolysis of extracellular ATP by the ectoenzymes CD39 and CD73[17] the expression of which is increased in hypoxic and immunosuppressive tissue environments[18]. Adenosine concentrations are therefore greatly enhanced in the tumor microenvironment[14,19]. Adenosine signals through a family of four adenosine receptors[14,15]. The adenosine 2A receptor (A$_{2A}$R) is highly expressed in T cells, whereas mRNA for the other three isoforms is barely or not at all detectable[20]. A$_{2A}$R blockade or T cell-specific deletion enhances anti-tumor immunity in many models, often with enhanced CTL tumor infiltration[21-24]. The localized generation of adenosine and its role in tumor infiltration make adenosine an attractive target as a regulator of tumor immunity with a direct focus on the tumor microenvironment.

Expression of the inhibitory receptor TIM3 increases with repeated T cell stimulation[25] reaching particularly high levels in tumors. High TIM3 expression in tumors is related to poor overall survival[26]. Blocking TIM3 can enhance anti-tumor immunity, in particular in combination with anti-PD-1 or chemotherapy[27]. While TIM3 is highly expressed on CD8+ TIL, TIM3 also regulates myeloid cell function, is highly expressed on CD4+ Tregs in tumors and TIM3 signaling can display features of costimulation, such as activation of Akt/mTOR[28,29]. Even though the mechanism of action of TIM3 thus remains unresolved, its preferential expression on CD8+ TIL makes it an attractive candidate for a direct regulator of the interaction between CTL and their tumor target cells.

Here we aim to determine whether effectors of tumor-mediated immune suppression can directly regulate the killing of tumor target cells by CTL. We complement an in vivo tumor model with a matching in vitro reconstruction of the tumor microenvironment based on the interaction of tumor spheroids with CTL in the absence of any other cell types[7,30,31]: Murine renal carcinoma cells expressing the hemagglutinin (HA) protein from influenza virus A/PR/8 as neo-tumor-specific antigen (RencaHA) are effectively recognized by the immune system in vivo and generate an immune-suppressive tumor microenvironment. Recognition of Renca tumors can be enhanced with the adoptive transfer of CL4 T cell receptor transgenic CTL recognizing an HA-derived peptide. In vitro, RencaHA cells cultured in three-dimensional spheroids effectively interact with CL4 CTL such that the suppressed phenotype acquired in this in vitro interaction closely matches that acquired by adoptively transferred CL4 CTL in vivo[7]. A key element of the mechanism of the impaired killing ability of tumor-suppressed CTL is the reduced ability of CTL to execute the cytoskeletal polarization steps required for effective target cell killing[7].

Blocking A$_{2A}$R partially reduced RencaHA tumor growth in vivo. Upon A$_{2A}$R blockade TIM3 was highly upregulated amongst CD8+ TIL. Combing in vivo blockade of A$_{2A}$R with that of TIM3 further reduced tumor growth in the context of the adoptive transfer of CL4 T cells. This combined treatment partially restored the ability of CTL to polarize towards their tumor target cells and enhanced tumor infiltration of CTL. In vitro, overexpression of TIM3 by CL4 CTL and treatment with an A$_{2A}$R agonist inhibited killing of tumor cells in tumor spheroids and the A$_{2A}$R agonist also reduced spheroid infiltration by CTL. An A$_{2A}$R agonist suppressed cytoskeletal polarization of CTL during both migration and coupling to tumor target cells. Interference with cytoskeletal polarization thus is a potential mechanism by which A$_{2A}$R and TIM3 may directly suppress the killing of tumor cells by CTL.

## Results

**An experimental approach to identify direct suppressors of CTL killing of tumor target cells.** Enhancing the ability of CTL to kill tumor target cells within the tumor microenvironment (TME) is of immense therapeutic interest. To identify immunosuppressive factors that act directly on CTL within the TME, we combined in vivo mouse studies with matched direct investigation of the interaction of CTL with tumor spheroids in vitro (Fig. 1)[7,30,31]: Renca renal carcinoma cells expressing influenza A/PR/8/H1N1 hemagglutinin (HA) induce an endogenous anti-tumor immune

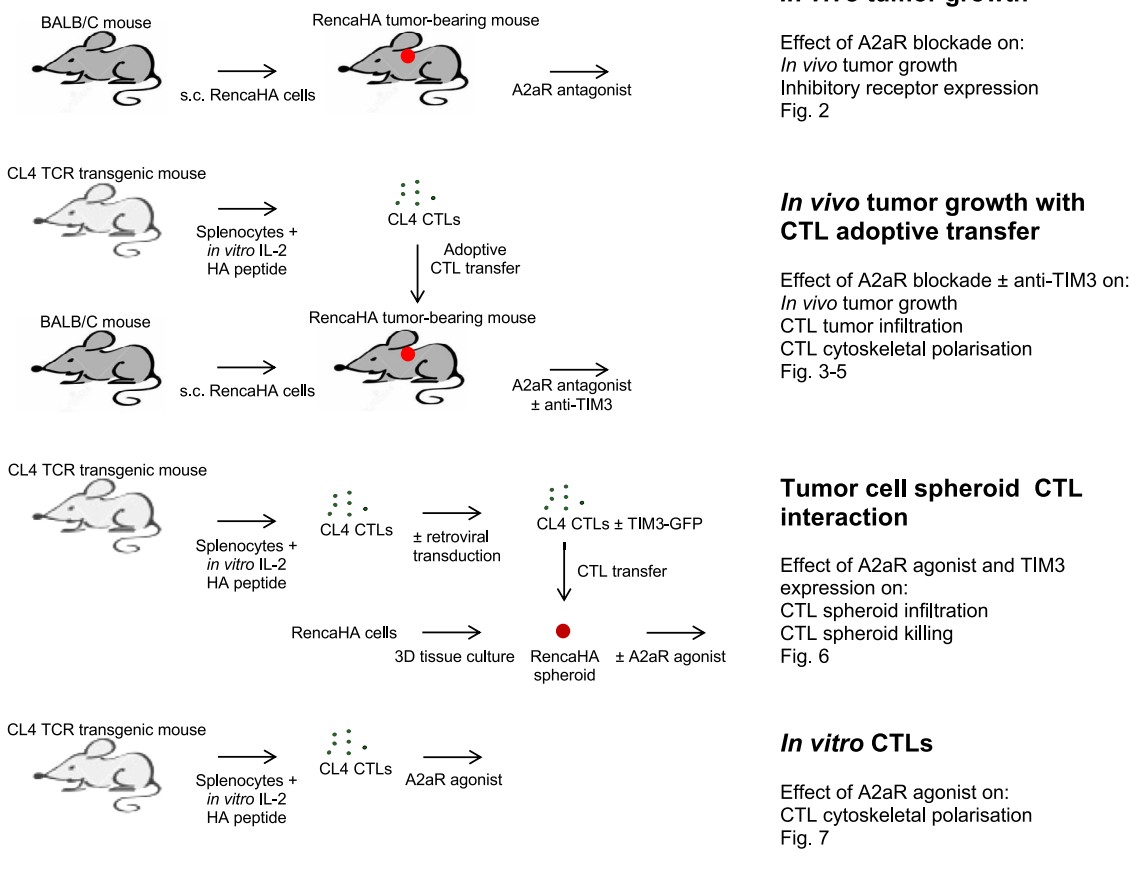

**Fig. 1 Schematic representation of the experimental system.** The different experimental strategies used are illustrated on the left with a list of corresponding experiments and figures shown for each on the right.

response and an immune-suppressive TME when grown subcutaneously in mice. The T cell receptor (TCR) of T cells from CL4 TCR transgenic mice recognizes the HA peptide 518–526 (IYST-VASSL) as restricted by H-2K$^d$. Upon adoptive transfer into RencaHA tumor-bearing mice, CL4 CTL infiltrates the tumor and acquires a suppressed phenotype[7]. Incubation of in vitro primed CL4 CTL with RencaHA tumor cells grown as three-dimensional spheroids induces a suppressed CTL phenotype that shares key features with tumor-infiltrating CL4 T cells[7]. Thus, we could characterize tumor-associated immunosuppression of CD8$^+$ T cells in parallel in the in vivo tumor model to establish physiological relevance and in the in vitro spheroids to establish direct effects on the interaction of CTL with tumor cells in the absence of other immune cells.

**The adenosine 2A receptor suppresses anti-tumor immunity.** High concentrations of adenosine occur within many solid tumors. CD4$^+$FoxP3$^+$ regulatory T cells (Tregs) express the adenosine-generating ectoenzymes CD39 and CD73 as an important means of in situ adenosine generation. Within TIL from RencaHA tumor-bearing mice 86 ± 2% of CD25$^+$FoxP3$^+$CD4$^+$ T cells expressed both CD39 and CD73 (Fig. 2a). CD25$^+$CD4$^+$ TIL from RencaHA tumor-bearing mice suppressed in vitro proliferation of naïve CL4 T cells in a manner dependent on the adenosine 2A receptor (A$_{2A}$R) (Supplementary Fig. 1a). This in vitro generation of functionally relevant amounts of adenosine by CD25$^+$FoxP3$^+$CD4$^+$ TIL suggests that these cells can also generate elevated adenosine concentrations in the RencaHA TME.

To determine whether A$_{2A}$R suppresses anti-tumor immunity in the RencaHA model, we treated RencaHA tumor-bearing mice

intraperitoneally with 10 mg/kg of the A$_{2A}$R antagonist ZM 241385 every 4 days (Fig. 1)[32,33]. Comparison with other emulsified compounds of a similar molecular weight suggests that such treatment led to a peak blood concentration of ZM 241385 in the low μM range with a half-life of about 1 h[34]. At that peak concentration, ZM 241385 inhibits both A$_{2A}$R and A$_{2B}$R[35]. However, A$_{2A}$R mRNA expression in T cells is several-fold higher than that of A$_{2B}$R; A$_{2A}$R-deficient lymphocytes do not elevate cAMP in response to adenosine any more[20] and ZM 241385 displays >50-fold selectivity for A$_{2A}$R over A$_{2B}$R as detailed in the methods section. For the remainder of the manuscript we, therefore, refer to ZM 241385 as an A$_{2A}$R antagonist. Under control conditions, tumors grew from 150 ± 25 mm$^3$ at day 12, the start of treatment, to 2075 ± 290 mm$^3$ within 8 days with two mice sacrificed early as their tumors exceeded maximum allowable tumor volume. ZM 241385 treatment resulted in a reduction in tumor growth from 140 ± 35 mm$^3$ to only 715 ± 85 mm$^3$ over 8 days with no mice needing to be sacrificed ($p < 0.01$) (Fig. 2b). These data establish a partial role of A$_{2A}$R in suppressing anti-tumor immunity in the RencaHA model.

**TIM3 expression is enhanced on CTL upon in vivo A$_{2A}$R blockade.** The only partial nature of the suppression of tumor growth upon A$_{2A}$R blockade suggests that other elements of tumor-mediated immune suppression may synergize with A$_{2A}$R engagement or are even upregulated to compensate for A$_{2A}$R blockade. We, therefore, determined the expression of inhibitory receptors: TIM3, TIGIT, LAG3, PD-1, and adenosine-producing ectoenzymes: CD39 and CD73 by both CD8$^+$ and CD4$^+$ TIL from ZM 241385-treated and control tumors.

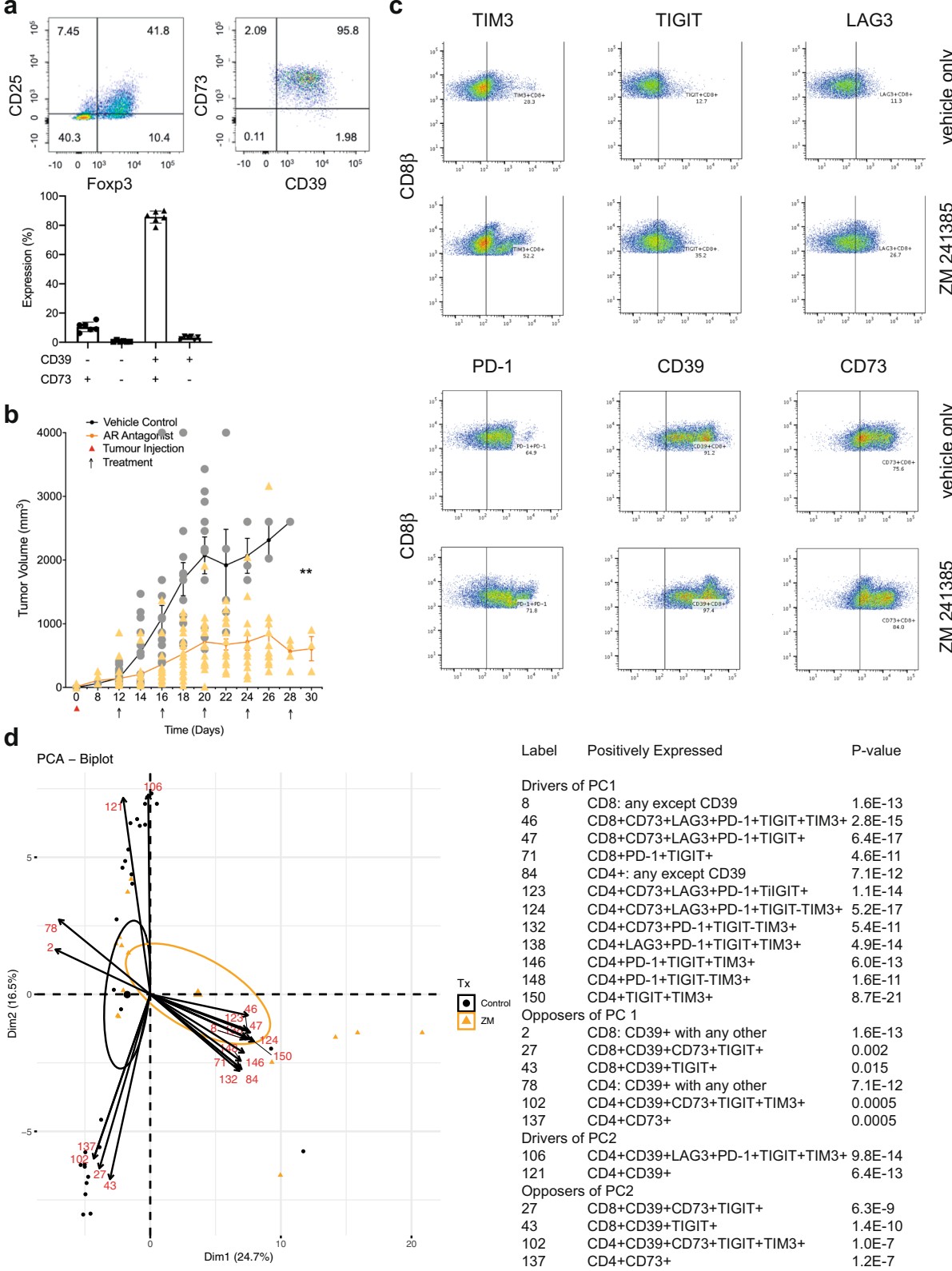

| Label | Positively Expressed | P-value |
|---|---|---|
| **Drivers of PC1** | | |
| 8 | CD8: any except CD39 | 1.6E-13 |
| 46 | CD8+CD73+LAG3+PD-1+TIGIT+TIM3+ | 2.8E-15 |
| 47 | CD8+CD73+LAG3+PD-1+TIGIT+ | 6.4E-17 |
| 71 | CD8+PD-1+TIGIT+ | 4.6E-11 |
| 84 | CD4+: any except CD39 | 7.1E-12 |
| 123 | CD4+CD73+LAG3+PD-1+TiIGIT+ | 1.1E-14 |
| 124 | CD4+CD73+LAG3+PD-1+TIGIT-TIM3+ | 5.2E-17 |
| 132 | CD4+CD73+PD-1+TIGIT-TIM3+ | 5.4E-11 |
| 138 | CD4+LAG3+PD-1+TIGIT+TIM3+ | 4.9E-14 |
| 146 | CD4+PD-1+TIGIT+TIM3+ | 6.0E-13 |
| 148 | CD4+PD-1+TIGIT-TIM3+ | 1.6E-11 |
| 150 | CD4+TIGIT+TIM3+ | 8.7E-21 |
| **Opposers of PC 1** | | |
| 2 | CD8: CD39+ with any other | 1.6E-13 |
| 27 | CD8+CD39+CD73+TIGIT+ | 0.002 |
| 43 | CD8+CD39+TIGIT+ | 0.015 |
| 78 | CD4: CD39+ with any other | 7.1E-12 |
| 102 | CD4+CD39+CD73+TIGIT+TIM3+ | 0.0005 |
| 137 | CD4+CD73+ | 0.0005 |
| **Drivers of PC2** | | |
| 106 | CD4+CD39+LAG3+PD-1+TIGIT+TIM3+ | 9.8E-14 |
| 121 | CD4+CD39+ | 6.4E-13 |
| **Opposers of PC2** | | |
| 27 | CD8+CD39+CD73+TIGIT+ | 6.3E-9 |
| 43 | CD8+CD39+TIGIT+ | 1.4E-10 |
| 102 | CD4+CD39+CD73+TIGIT+TIM3+ | 1.0E-7 |
| 137 | CD4+CD73+ | 1.2E-7 |

To identify combinations of inhibitory receptor expression altered upon $A_{2A}R$ blockade, we used a principal component analysis. Input data were the percentage of T cells expressing inhibitory receptors in all combinations and tumor size (Fig. 2c, d, Supplementary Fig. 1b). Principal component (PC) 1 effectively distinguished TIL from ZM 241385-treated and control mice (95% confidence ellipses are shown). Variables that contribute to

PC1 were therefore positively associated with $A_{2A}R$ antagonism of TIL. TIM3 expression was upregulated in combination with other inhibitory receptors by $A_{2A}R$ antagonist treatment and contributed to PC1 with $p < 0.001$. PD-1 and TIGIT expression were also strongly associated with PC1, albeit TIGIT both amongst drivers and opposers. Thus, upregulation of the expression of PD-1 and TIM3 is most strongly associated with

**Fig. 2 An $A_{2A}R$ antagonist delays in vivo tumor growth and triggers compensatory upregulation of T cell inhibitory receptors. a** TIL from RencaHA tumor-bearing BALB/c mice was stained with anti-CD25, anti-FoxP3, anti-CD39, and anti-CD73 mAb. On the left and in the middle, representative flow cytometry data are shown. On the right, percentage TIL expressing CD39 and CD73 are given as mean ± SEM for $N = 6$ experiments. **b** Mean RencaHA tumor volume is given ± SEM in BALB/c mice after s.c. injection of $1 \times 10^6$ RencaHA tumor cells on day 0 and i.p. injection with ZM 241385 when tumors had reached 5 mm diameter in any one direction (day 12–14) and further treatments administered every other day as indicated (linear mixed model to perform repeated measures ANOVA. **$p < 0.01$). $N = 26$ treated mice and 20 control mice over four separate experiments. **c** CD45$^+$ cells from the RencaHA tumor-bearing mice treated with ZM 241385 or vehicle control in **b** were stained using mAbs against CD8, CD4, CD39, CD73, TIM3, TIGIT, LAG3, and PD-1. Representative flow cytometry data are shown. The gating strategy for the identification of CD8$^+$ TIL is given in Supplementary Fig. 1b. **d** The outcome of a principal component analysis is given with input data of percentage expression of markers in **c** in each combination of the eight markers, representing 308 variables, and tumor volume as an additional variable. Each triangle (ZM 241385-treated) or circle (control-treated) represents an individual tumor-bearing mouse. Large symbols represent the average position of treated and control mice along PC1 with ellipses showing 95% confidence intervals. The 24 variables making the greatest contribution to principal component (PC) 1 and 2 are overlaid as numbered vectors and are listed in the table. Entire FACS data are available as detailed in the data availability statement. Source data are provided in Supplementary Data 1.

$A_{2A}R$ blockade. As TIM3 is more likely to directly affect the interaction of CTL with their tumor cell targets as discussed in the introduction, we selected TIM3 blockade as an adjunct treatment to improve responses to $A_{2A}R$-antagonism. Expression of various combinations of inhibitory receptors by CD4$^+$ TIL was also associated with $A_{2A}R$ antagonist treatment, as not further pursued here because of our focus on the direct interaction between CD8$^+$ CTL and tumor target cells.

PC2 separated TIL by the volume of the tumor they are derived from, with larger tumors associating positively with PC2. This highlights the importance of accounting for tumor volume when assessing immune profiles. CD39 expression strongly drove PC2 while opposing PC1. This indicates that with increasing size control tumors relied more on adenosine for immune suppression as opposed by $A_{2A}R$ antagonism. Only in the largest tumors, >1,000 mm$^3$, did inhibitory receptor expression in TIL start to increase as an additional suppressive mechanism (Supplementary Fig. 1c). Conversely, amongst $A_{2A}R$-blocked tumors, TIL inhibitory receptor expression was already high in smaller tumors, providing an alternate means of suppression in the absence of $A_{2A}R$ (Supplementary Fig. 1c). Together these data suggest that RencaHA tumors rely strongly on $A_{2A}R$ for immune suppression with compensatory upregulation of inhibitory receptor expression upon $A_{2A}R$ blockade.

**TIM3 enhances suppression of CD8$^+$ T cell-dependent antitumor immunity by $A_{2A}R$.** To investigate immunosuppressive synergy between $A_{2A}R$ and TIM3, groups of BALB/c mice bearing 12-day old RencaHA tumors were treated with the $A_{2A}R$ antagonist ZM 241385 plus or minus the anti-TIM3 blocking mAb RMT3-23 (Fig. 1). For precise temporal synchronization and a standardized number of anti-tumor CTL, we used i.v. double adoptive T cell transfer (ATT) of $5 \times 10^6$ CL4 TCR transgenic CTL on days 12 and 14. We observed three phases of tumor growth upon treatment, an initial 'growth' phase of 6 days, a 'response' phase of 8 days, and a subsequent open-ended 'relapse' phase (Fig. 3a). As an overall outcome measure, we determined a ratio of the final tumor volume after growth, response, and relapse to initial tumor volume at the start of treatment. Adoptive transfer of CL4 CTL plus both ZM 241385 and anti-TIM3 mAb gave a significant ($p < 0.05$) reduction in the average final to initial tumor volume ratio compared with mice receiving adoptive CL4 T cell transfer plus ZM 241385 only. TIM3 thus enhanced suppression of anti-RencaHA tumor immunity by $A_{2A}R$ (Fig. 3b).

To determine at which stage of tumor growth treatment with ZM 241385 and anti-TIM3 mAb exerted its effect, we analyzed the three phases of tumor growth separately. Initial tumor growth in all mice peaked between day 14–18. The combination of adoptive transfer of CL4 CTL with both ZM 241385 and anti-

TIM3 mAb resulted in a marked reduction in tumor growth relative to mice given either CL4 T-cells alone or CL4 CTL plus ZM 241385 (Fig. 3c) suggesting that TIM3 enhances suppression of anti-RencaHA tumor immunity by $A_{2A}R$ already in this early phase of tumor growth.

In the 'response' phase, between days 20 and 26, most tumors regressed in all treatment groups. Comparing this regression to the continued RencaHA tumor growth in the absence of CL4 CTL adoptive transfer (Figs. 2b, 3a), the regression can be attributed to the transferred CL4 CTL. Mice in which tumors regressed were categorized as 'responders'. The percentage of responder mice did not differ between the treatment groups (Fig. 3d). We then defined 'relapse' as tumors reaching a volume ≥ the starting volume. When analyzing all mice, relapse was slightly delayed in mice treated with ZM 241385 and anti-TIM3 mAb without reaching statistical significance (Fig. 3e). When analyzing relapse only in responders, relapse frequencies differed substantially with treatment. Upon adoptive transfer of CL4 CTL alone, 8/17 tumors relapsed within 26 days of regression. Upon treatment with adoptive transfer of CL4 CTL and ZM 241385, tumor relapse occurred in 8/10 responder mice. However, when anti-TIM3 mAb was used in combination with ZM 241385 plus adoptive transfer of CL4 CTL, only 3/12 mice underwent relapse after regression ($p < 0.05$ compared to treatment with CL4 adoptive transfer plus ZM 241385). Thus, blocking $A_{2A}R$ together with TIM3 decreased RencaHA tumor relapse.

To elucidate mechanisms of the prevention of tumor relapse, we determined T cell persistence. After the initial response phase adoptively transferred Thy1.1$^+$ CL4 CTL were only detected in mice from the ZM 241385 plus anti-TIM3 mAb group. They could also be expanded by a booster immunization with the HA peptide in mice receiving only CL4 CTL (Fig. 4a, Supplementary Fig. 2a). However, Thy1.1$^+$ CL4 CTL were not detected in the group treated with ZM 241385 alone, with or without a booster immunization, consistent with the high relapse rate of 8/10. To determine the role of persistent CD8$^+$ T cells in tumor immunity, we depleted them. In mice with complete and durable tumor remission lasting >8 days, only five out of 19 mice experienced relapsed (Fig. 4b). In addition, responder mice were resistant to tumor growth following rechallenge with tumor cells (Fig. 4c). In contrast, treatment with anti-Thy1.1 depleting mAb at 28 days induced tumor re-growth in 4/4 mice (Fig. 4d). Treatment with anti-CD8β depleting mAb resulted in tumor re-growth in 2/2 mice (Fig. 4e, Supplementary Fig. 2b) ($p < 0.01$ of combined depletion data versus control). Together, these data establish that response and suppression of relapse depend on the continued presence of tumor-reactive CTL.

Thus far, our in vivo data establish that combined treatment with ZM 241385 and anti-TIM3 mAb enhances anti-tumor immunity

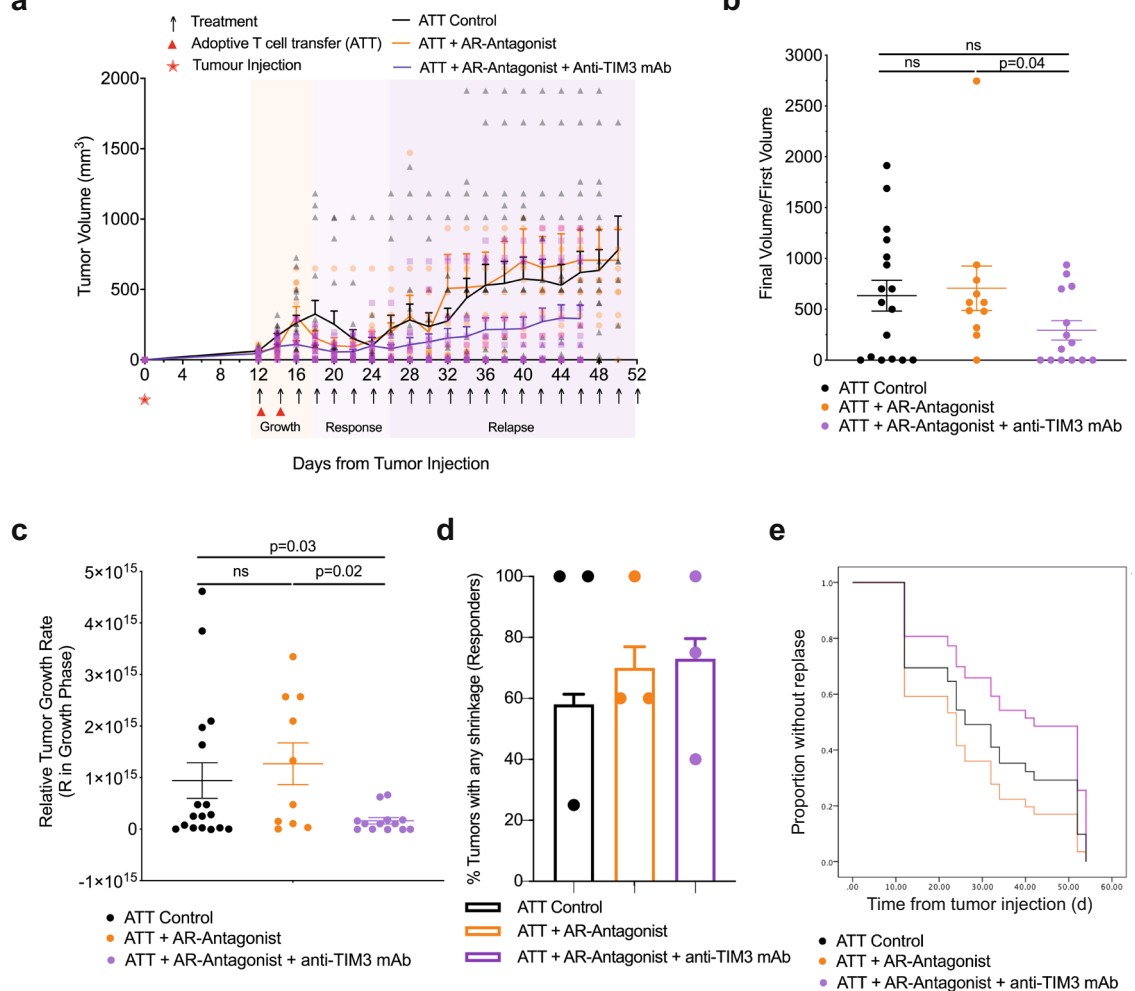

**Fig. 3 A_2AR antagonist and anti-TIM3 mAb synergistically suppress RencaHA tumor growth. a** RencaHA tumor-bearing BALB/c mice were injected i.v. twice with $5 \times 10^6$ purified CL4 CTL on days 12 and 14 and treated with A_2AR-antagonist, anti-TIM3 mAb, or vehicle + isotype-control antibody, as shown. Tumor growth is displayed as mean tumor volumes + SEM with $N = 42$ mice over three independent experiments, with at least 11 mice per group. **b** The ratio of final to initial (day 12) tumor volume of tumors in a is given with the mean ± SEM (independent two-sample $t$-test). **c** $R$-values representing the growth rate of tumors in a, between 12 and 16 days, are given with the mean ± SEM (independent two-sample $t$-test). **d** Proportion of tumors across all experimental replicates in a which were regressed are given as bars ±SEM with proportion in individual experimental replicates as dots. Neither analysis of pooled data using Fisher's exact Boschloo test nor of experimental replicates using 1-way ANOVA yield significant differences. **e** Progression free survival of mice in a is given. Cox Proportional Hazards Regression analysis. Treatment did not significantly predict progression free survival: ($p = 0.24$ ATT control; $p = 0.40$ ATT + A_2AR-Antagonist; $p = 0.30$ ATT + A_2AR-Antagonist + Anti-TIM3 mAb). Hazard ratio of progression versus Control was as follows: ATT + A_2AR-Antagonist = 0.36 (0.61 ± 3.38), ATT + A_2AR-Antagonist + Anti-TIM3 mAb = −0.53 (0.21 ± 1.63). Source data are provided in Supplementary Data 2.

in a T cell-dependent fashion. The mechanism underpinning the immune-enhancing effect of such treatment in the initial anti-tumor response is the focus of the remainder of this manuscript.

**A_2AR and TIM3 suppress the cellular polarization of CTL and tumor infiltration.** Mechanisms of A_2AR and TIM3 in suppression of anti-tumor immunity are of interest. For effective tumor infiltration and tumor cell killing CTL need to undergo a series of cytoskeletal polarization steps. The inability to effectively execute such steps characterizes tumor-infiltrating CTL[7]. To determine if A_2AR and TIM3 regulate effective TIL polarization in the tumor microenvironment, we isolated TIL from RencaHA tumors and imaged their subsequent interaction with K^dHA peptide pulsed Renca cells ex vivo. One hallmark of defective cytoskeletal TIL polarization is CTL lamellae directed away from the cellular interface with the tumor target cell to destabilize the cell couple

(Fig. 5a). Less than 30% of in vitro primed control CL4 CTL display such lamellae even after 15 min of cell contact; whereas, all TIL from control-treated tumors do so within 7 min (Fig. 5b). Following treatment of mice with the A_2AR antagonist ZM 241385 plus blocking anti-TIM3 mAb off-interface lamellae formed later, with the median time of first off-interface lamellae formation delayed from 100 s to 220 s. 11% of cell couples did not show any off-interface lamellae at all (Fig. 5b). Another hallmark of defective TIL cytoskeletal polarization is T cell translocation over the tumor cell surface away from the site of initial coupling (Fig. 5c). While almost completely absent in in vitro primed control CTL, 71 ± 9% of TIL displayed such translocation (Fig. 5d, e). This frequency was significantly ($p < 0.001$) reduced to 37 ± 9% and 23 ± 7% upon tumor treatment with ZM 241385 alone or in combination with anti-TIM3 mAb, respectively. Data from these restoration experiments establish that A_2AR, as enhanced by TIM3, contributes to the

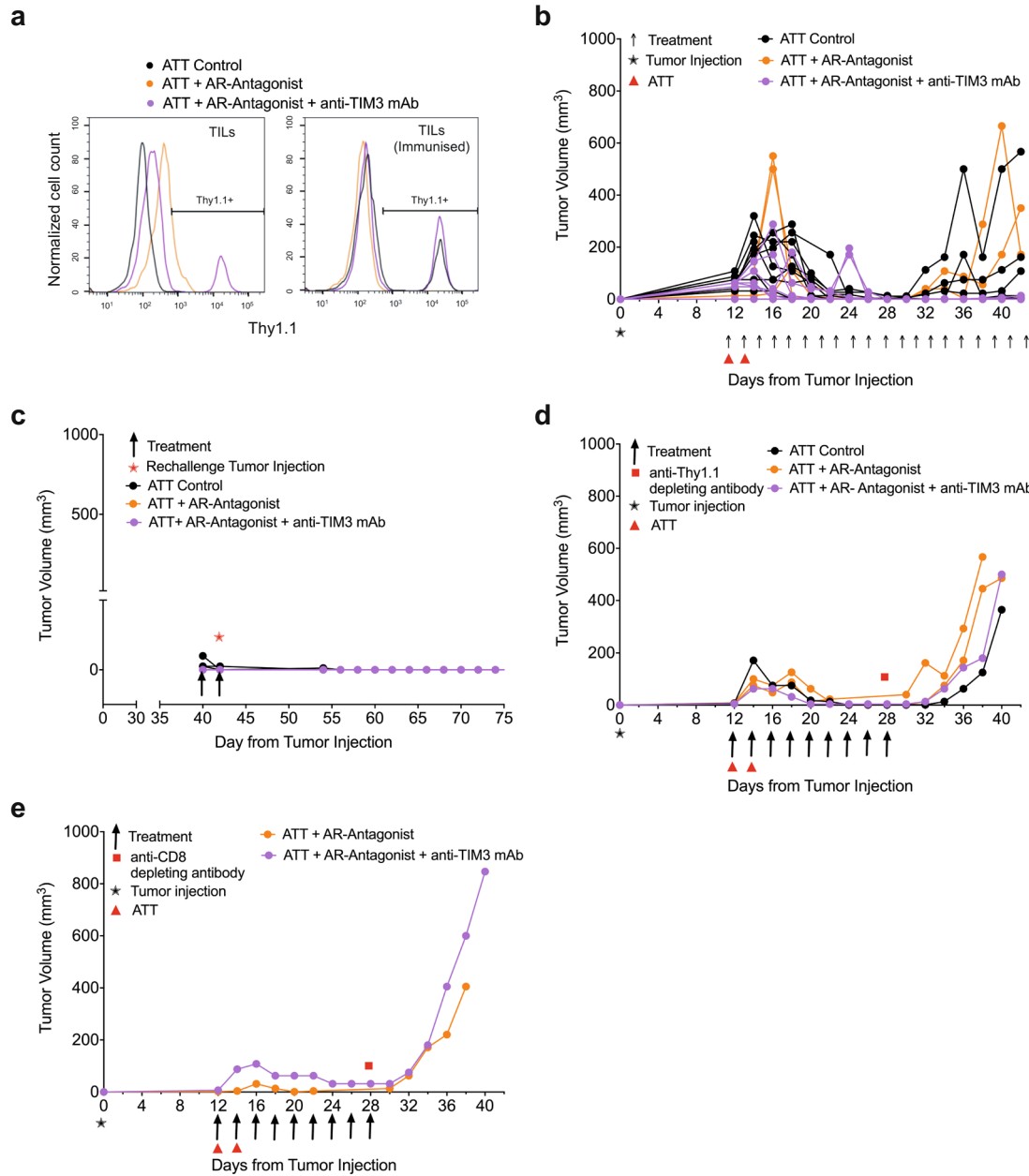

**Fig. 4 A₂ₐR antagonist plus anti-TIM3 mAb diminish tumor relapse in a T cell-dependent fashion.** Mouse numbers across the different parts of the experiments are detailed in Supplementary Fig. 2a. **a** RencaHA tumor-bearing BALB/c were injected i.v. with two doses of $5 \times 10^6$ Thy1.1$^+$ CL4 CTL (ATT) plus either vehicle and isotype control, A₂ₐR-antagonist (ZM 241385) alone or A₂ₐR antagonist + anti-TIM3 mAb as shown. The percentage of Thy1.1$^+$ CL4 TIL between days 25 and 48 is given as representative data from $N = 25$ mice over three independent experiments. Mice were either immunized with HA peptide 5 days prior to TIL harvest ($N = 5$) to induce expansion of Thy1.1 + CL4 T cells or immunized with empty vehicle control ($N = 19$); one representative graph is shown for each group. **b–e** Tumor growth curves from individual mice treated in Fig. 3a, which had experienced complete and durable tumor remission. **b** untreated and **c** rechallenged with tumor cells at day 40; **d** depleted of Thy1.1$^+$ T cells using anti-Thy1.1 depleting mAb at 28 days; **e** depleted of all CD8$^+$ T cells using anti-CD8β depleting mAb at 28 days with depletion efficiency shown in Supplementary Fig. 2b. Source data are provided in Supplementary Data 3.

defective cytoskeletal polarization of TIL. Consistent with the importance of such CTL cytoskeletal polarization for cytolysis, treatment of tumors with the A₂ₐR antagonist ZM 241385 plus blocking anti-TIM3 mAb significantly ($p = 0.02$) enhanced ex vivo CL4 TIL killing (Fig. 5f, g). We have also seen the restoration of CTL cytoskeletal polarization and killing upon treatment of RencaHA tumor-bearing mice with anti-PD-1[7]. It is now of interest whether or not A₂ₐR and TIM3, in contrast to PD-1[7], directly regulate the interaction between CTL and their tumor target cells.

As an additional process requiring cytoskeletal activity, we examined the infiltration of endogenous CD8$^+$ T cells and CL4 CTL into Renca tumors in vivo. Such infiltration was focused on peripheral tumor regions and was enhanced upon treatment with the A₂ₐR antagonist ZM 241385 plus blocking anti-TIM3 mAb (Fig. 5h, i). In contrast, CD4$^+$ Treg infiltration was diminished (Fig. 5i). An enhanced ratio of CD8$^+$ CTL to CD4$^+$ Tregs in the tumor thus constitutes an additional potential mechanistic contribution of reduced tumor growth in mice upon blockade of the A₂ₐR and TIM3.

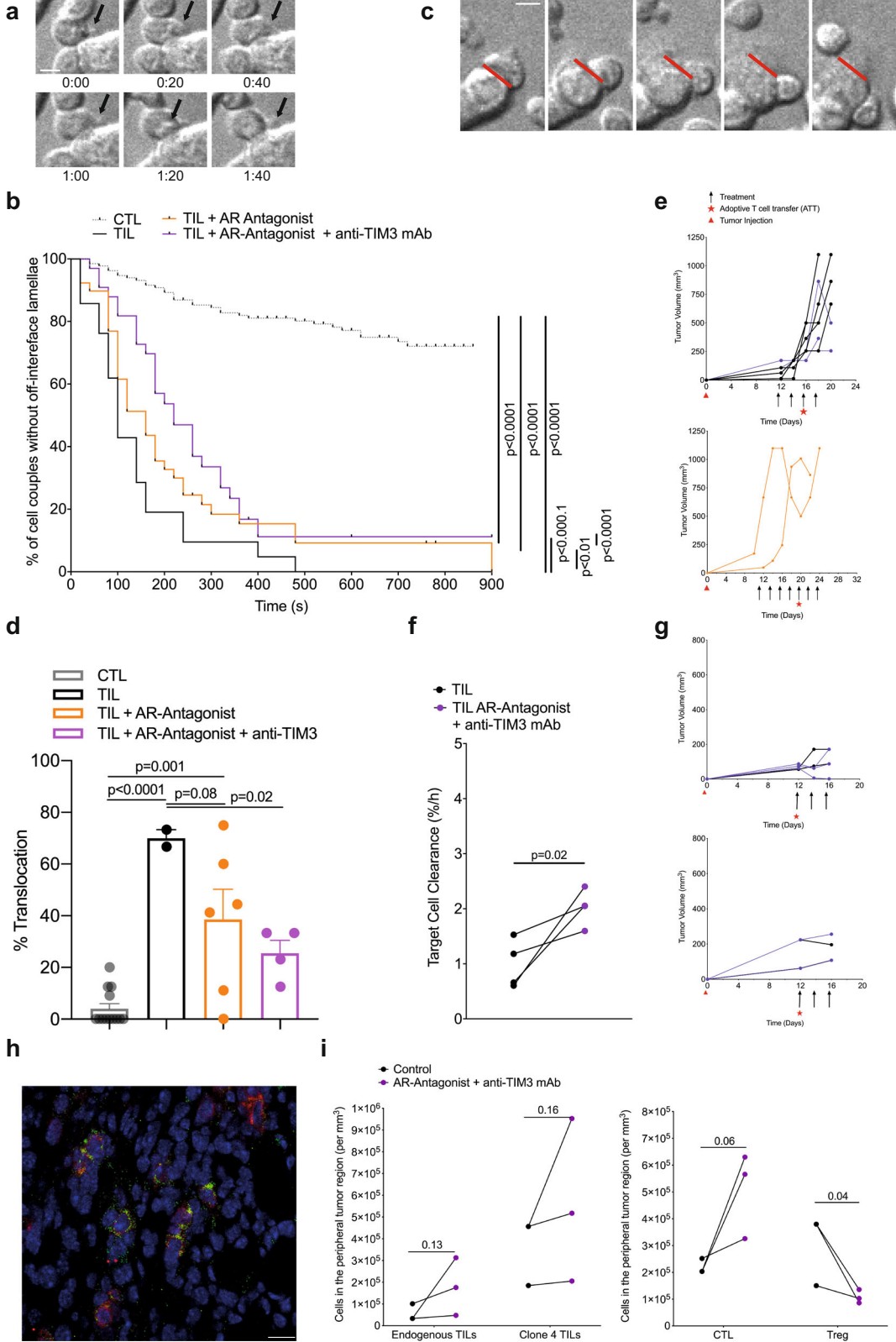

## A$_{2A}$R and TIM3 directly inhibit the killing of tumor target cells by CL4 CTL in spheroids

**A$_{2A}$R and TIM3 directly inhibit the killing of tumor target cells by CL4 CTL in spheroids**. To determine whether A$_{2A}$R and TIM3 directly regulate the interaction of suppressed CTL with their tumor target cells, we employed the in vitro reconstruction of CL4 CTL suppression in our Renca spheroid/CL4 CTL only system (Fig. 1). To determine the roles of TIM3, we overexpressed a TIM3-GFP fusion protein in CL4 CTL. Such overexpression did

not diminish spheroid infiltration by CL4 CTL (Fig. 6a). However, it diminished the ability of CL4 CTL to kill Renca target cells inside the spheroids as determined with DRAQ7 staining (Fig. 6b). Confirming TIM3-dependence, diminished Renca tumor cell killing could be reversed with the anti-TIM3 blocking mAb RMT3-23 (Fig. 6b). To determine the roles of A$_{2A}$R in tumor cell killing by CTL, we treated CL4 CTL/Renca spheroid

**Fig. 5 $A_{2A}R$ plus TIM3 suppress CTL polarization, killing, and infiltration in the TME. a–e** Ex vivo cytoskeletal polarization of CL4 TIL from RencaHA tumor-bearing BALB/c mice upon ATT of CL4 T cells and treatment with combinations of $A_{2A}R$ antagonist and anti-TIM3 mAb ($N = 5$ control, $N = 3$ both treatments, $N = 2$ $A_{2A}R$ antagonist alone) in comparison to in vitro CL4 CTL is given. Mouse numbers across the different parts of the experiments are detailed in Supplementary Fig. 2a. **a** Representative image sequence in 20 s steps of a CL4 CTL with off-interface lamellae as labeled with black arrows. Time 0:00 is arbitrary. Scale bar = 5 μm. Entire image sequence in Supplementary Movie 1 with a second time series in Supplementary Movie 2. **b** Time until the formation of the first off-interface lamella is given. Kaplan–Meier survival analysis (Log Rank) $N > 30$ cell couples per condition over four experiments. **c** Representative image sequence in 1 min steps of a CL4 CTL with a translocation, i.e. CTL movement of >1 interface diameter from the initial location of the immune synapse (red line). Time 0:00 is the time of tight cell coupling. Scale bar = 5 μm. Entire image sequence in Supplementary Movie 3 with a second time series in Supplementary Movie 4. **d** The frequency of cell couples with translocation is given ± SEM. 1-way ANOVA. $N = 21$–132 cell couples per condition over ≥2 experiments. Alternate analysis of pooled data of percent translocation by proportion's $z$-test yields $p < 0.0001$ for all comparisons between CTL/TIL and treated TIL samples. **e** Growth curves of the tumors used for analyses **a**–**d** are shown. **f** RencaHA tumor-bearing BALB/c mice were treated with $A_{2A}R$ antagonist and anti-TIM3 mAb from day 12. ATT of CL4 T cells was given on day 12. The killing of $K^dHA$-pulsed Renca mCherry tumor cells by day 16 ex vivo CL4 TIL is given at an E:T ratio of 3:2. Each point = 1 tumor, $N = 2$ experiments. Size-matched tumors analyzed on the same day are paired for comparison using a $t$-test. **g** Growth of the tumors used for killing analysis in **f** are shown as two separate experiments. **h, i** Half of the size-matched tumors from mice in e and g were stained. **h** A representative image with CD8 staining in red and Thy1.1 staining in green. scale bar = 50 μm. **i** The numbers of (left panel) endogenous $Thy1.2^+CD8^+$ TIL and adoptively transferred $Thy1.1^+$ Clone 4 TIL or (right panel) total $CD8^+$ TIL and $FOXP3^+$ regulatory T cells in ten peripheral and ten central tumor areas are given. $N = 2$ control tumors and three treated tumors analyzed over two experiments. Size-matched tumors fixed on the same day are paired for analysis using $a$ $t$-test. Source data are provided in Supplementary Data 4.

co-cultures with the $A_{2A}R$ agonist CGS-21680 at 1 μM[36] (Fig. 6c–e). Such treatment led to a significant ($p < 0.05$) reduction of CL4 CTL infiltration into the spheroids to about half of the level of infiltration seen with the vehicle only control (Fig. 6c, e), consistent with previously described enhancement of CTL infiltration into tumors upon $A_{2A}R$ antagonist treatment[24] and in $A_{2A}R$-deficient mice[22]. In CGS-21680 treated spheroids, Renca cell death was drastically diminished reaching only 16% of control at the 12 h time point ($p < 0.005$) (Fig. 6d, e). As the effect of CGS-21680 on killing is substantially greater than the effect on infiltration, reduced spheroid infiltration can only partially account for reduced tumor cell death upon treatment with the $A_{2A}R$ agonist. Therefore, $A_{2A}R$ likely also impairs the execution of tumor cell killing. In combination, the spheroid data establish that $A_{2A}R$ and TIM3 directly suppress the ability of CL4 T cells to kill tumor target cells with an additional inhibitory effect of $A_{2A}R$ in reducing CTL infiltration.

**Adenosine impairs CTL cytoskeletal polarization.** Restoration of cytoskeletal polarization and increased tumor infiltration were potential mechanisms of immune enhancement in vivo in TIL from tumors treated with the $A_{2A}R$ antagonist ZM 241385 together with anti-TIM3 mAb. To determine the direct roles of $A_{2A}R$ in cytoskeletal polarization and CTL effector function, we investigated CL4 CTL function in vitro (Fig. 1). Migratory T cells extend a leading lamella and form a uropod at the posterior end. CL4 CTL treatment with the $A_{2A}R$ agonist CGS-21680 at 1 μM reduced the percentage of CL4 T cells with a uropod from $80 \pm 4\%$ in control-treated CL4 CTL to $25 \pm 5\%$ ($p < 0.001$) (Fig. 7a), indicative of suppression of the migratory phenotype. These data are consistent with the suppression of CL4 CTL spheroid infiltration upon treatment with CGS-21680 (Fig. 6c). Formation of a tight cell couple of a CTL with a tumor cell is the first step of killing and requires the effective extension of lamellae towards the target cell as a cytoskeletal polarization step. Upon treatment with CGS-21680 the frequency of CL4 CTL forming a tight cell couple upon contact with RencaHA target cells was significantly ($p < 0.01$) reduced from $49 \pm 6\%$ upon control treatment to $32 \pm 7\%$ (Fig. 7b). In CTL tumor cell couples, T cell translocation over the tumor target cell surface and off-interface lamellae are defining cytoskeletal features of suppressed TIL. Treatment of CL4 CTL with 1 μM or 10 μM of the pan-adenosine receptor agonist NECA enhanced the percentage of CL4 CTL that translocated over the tumor target cell surface from $4 \pm 2\%$ to

$36 \pm 9\%$ and $29 \pm 4\%$, respectively ($p < 0.001$) (Fig. 7c). This enhancement was reversed by parallel treatment with ZM 241385 at 1.25 μM. NECA and ZM 241385 at the concentrations used engage both $A_{2A}R$ and $A_{2B}R$[35]. However, as $A_{2A}R$ mRNA expression in T cells is several-fold higher than that of $A_{2B}R$ and $A_{2A}R$-deficient lymphocytes don't elevate cAMP in response to adenosine any more[20], the induction of CL4 T cell translocation and the reversion thereof are most likely mediated by $A_{2A}R$. Similarly, off-interface lamellae became more frequent upon CL4 CTL treatment with NECA and formed more rapidly, as both partially reversed by parallel treatment with ZM 241385 (Fig. 7d). Together these data establish that engagement of $A_{2A}R$ on CTL suppresses cytoskeletal polarization at multiple stages of CTL function, migration, cell couple formation and the maintenance of a fully polarized cell couple. These data corroborate a cytoskeletal mechanism for $A_{2A}R$-mediated suppression of TIL function. They also constitute an interesting contrast to PD-1, as investigated in the same experimental system[7]. As CTL constitutively express PD-1 and Renca cells PD-L1, we used anti-PD-1 to investigate the role of PD-1 in cytoskeletal polarization. Rather than restoring CL4 CTL and ex vivo TIL cytoskeletal polarization as expected for blocking an inhibitory interaction, anti-PD-1 increased the frequency of occurrence of off-interface lamellae and the translocation phenotype[7]. In addition, calcium signaling was impaired not enhanced[7]. The inhibition of cytoskeletal polarization in the direct interaction of CTL with their target cells by $A_{2A}R$ thus is in contrast to the stimulatory role of PD-1.

Cytoskeletal polarization also contributes to CTL signaling and effector function other than migration and killing. We therefore investigated CTL calcium signaling and naïve T cell proliferation. The elevation of the cytoplasmic calcium concentration in CL4 CTL upon interaction with $K^dHA$ peptide pulsed Renca tumor target cells was partially inhibited by treatment with 1 μM CGS-21680 (Fig. 7e). Proliferation of naïve CL4 T cells upon stimulation with anti-CD3/CD28 was partially inhibited by 1 μM or 20 μM NECA, as reversed with parallel treatment with the $A_{2A}R$ antagonist ZM 241385 at 1.25 μM without reaching statistical significance (Fig. 7f). An only moderate effect of $A_{2A}R$ on T cell proliferation has been previously reported[19]. A determination of whether these defects are secondary to impaired cytoskeletal polarization or independent thereof is beyond the scope of this manuscript. Such defects, albeit moderate in size, may contribute to the immune-suppressive function of $A_{2A}R$.

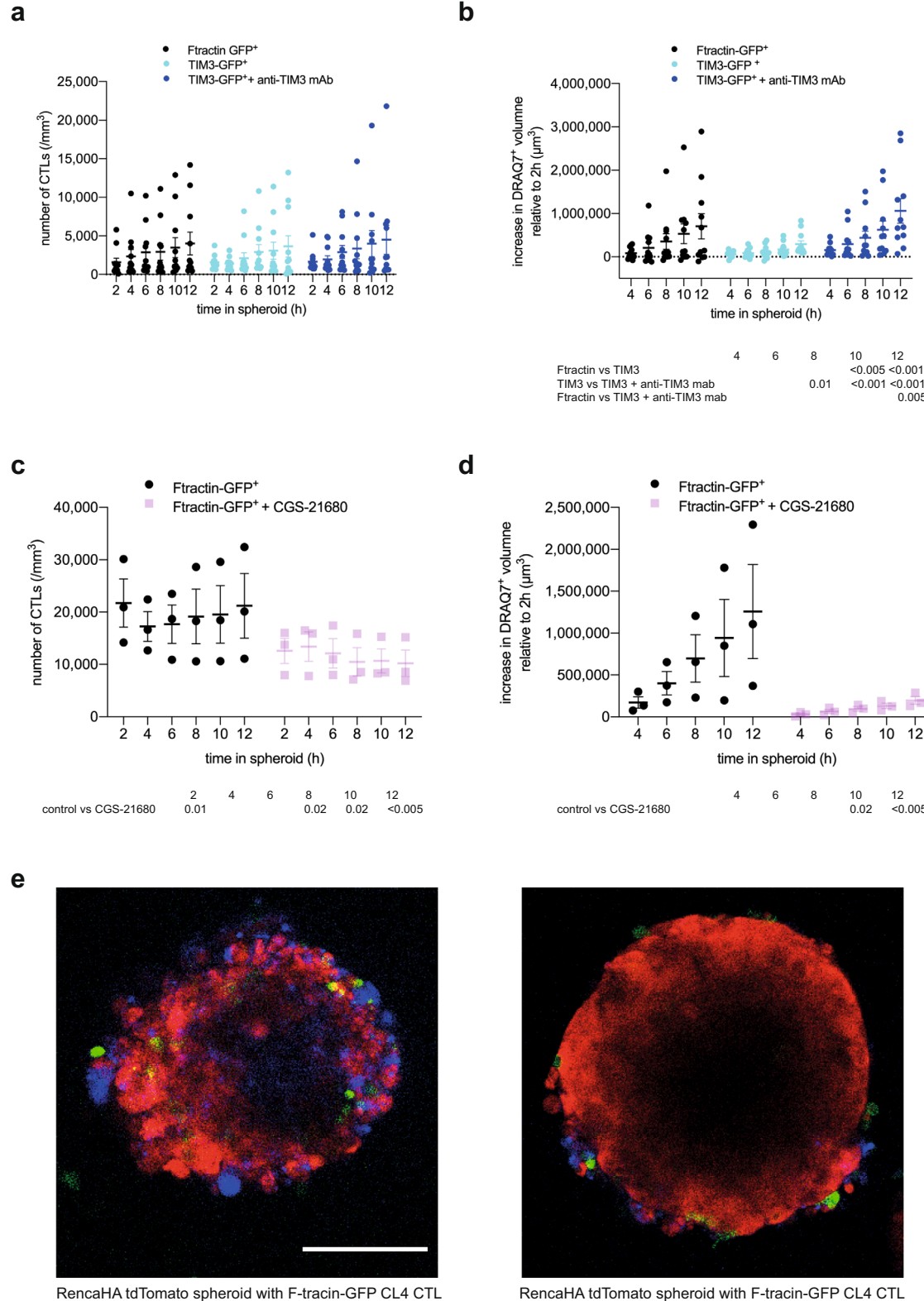

**a**

**b**

| | 4 | 6 | 8 | 10 | 12 |
|---|---|---|---|---|---|
| Ftractin vs TIM3 | | | | <0.005 | <0.001 |
| TIM3 vs TIM3 + anti-TIM3 mab | | | 0.01 | <0.001 | <0.001 |
| Ftractin vs TIM3 + anti-TIM3 mab | | | | | 0.005 |

**c**

**d**

| | 2 | 4 | 6 | 8 | 10 | 12 |
|---|---|---|---|---|---|---|
| control vs CGS-21680 | 0.01 | | | 0.02 | 0.02 | <0.005 |

| | 4 | 6 | 8 | 10 | 12 |
|---|---|---|---|---|---|
| control vs CGS-21680 | | | | 0.02 | <0.005 |

**e**

RencaHA tdTomato spheroid with F-tracin-GFP CL4 CTL

RencaHA tdTomato spheroid with F-tracin-GFP CL4 CTL + CGS-21680

## Discussion

To build a diverse tool kit of reagents for comprehensive cancer immunotherapy, it is vital we understand mechanisms of action of mediators of immune suppression. Using a matched in vivo, ex vivo, and in vitro spheroid approach, we have established that $A_{2A}R$ and TIM3 directly suppress the ability of CTL to kill tumor target cells. However, such establishment of a direct effect does not rule out the existence of additional indirect effects by hich $A_{2A}R$ and TIM3 blockade could enhance tumor immunity. For example, TIM3 is expressed on regulatory T cells in the tumor microenvironment across many tumor models, including Renca[37]. Blocking TIM3 diminishes the suppressive function of such regulatory T cells and thus promotes anti-tumor immunity[37]. Our data consistently link defects in CTL

**Fig. 6 $A_{2A}R$ and TIM3 suppress infiltration and killing of tumor spheroids by CTL. a, b** CTL retrovirally transduced to express TIM3-GFP or F-tractin-GFP as control were cocultured with RencaHA tdTomato spheroids incubated with $K^dHA$ peptide for 12 h ± 10 µg/ml anti-TIM3 mAb (clone RMT3-23) with images acquired every 2 h. Each data point is an independent experiment (N = 11) with five or six spheroids analyzed per independent experiment. **a** SIL densities are shown with the mean ± SEM. **b** Spheroid death, as measured by the increase in DRAQ7+ spheroid volume, is shown for the same experiments as in **a**. **c, d** CL4 CTL transduced to express F-tractin-GFP were cocultured with RencaHA tdTomato spheroids ±1 µM CGS-21680. Each data point is an independent experiment (N = 3) with three spheroids analyzed per independent experiment. **c** CL4 T cell densities are shown as mean ± SEM. **d** Spheroid death, as measured by the increase in DRAQ7+ spheroid volume, is shown for the same experiments as in **c**. All data were analyzed by 2-Way ANOVA, matched by independent repeat and time point. **e** Representative images of RencaHA tdTomato spheroids (red) with F-tractin-GFP-expressing CL4 CTL (green) as stained for cell death with DRAQ7 (blue) ± 1 µM CGS-21680 as indicated. scale bar = 100 µm with the same scale for both images. Entire imaging data are available as detailed in the data availability statement. Source data are provided in Supplementary Data 5.

cytoskeletal polarization to diminished killing. However, while diminished killing is a plausible explanation for the in vivo effects of TIM3 and $A_{2A}R$, only by direct in vivo manipulation of cytoskeletal polarization, in future experiments, one can prove that such impaired polarization limits tumor immunity. The engagement of a TCR by its physiological cognate MHC/peptide ligand is critical for the investigation of cytoskeletal regulation[38]. To allow such TCR engagement, the experiments here were conducted in a murine system. Previous work on adenosine and TIM3 suggests that findings are likely to be directly applicable in humans. For example, the preclinical efficacy of $A_{2A}R$ antagonists which are now licensed for control of Parkinson's disease was initially established in mice[39,40]. Elevated expression of TIM3 on Th1 cells is observed in both multiple sclerosis and its established mouse model of experimental autoimmune encephalitis and is required for the suppression of autoimmunity[41]. Moreover, the blocking anti-TIM3 antibody used here targets the same phosphatidylserine-binding site of TIM3 as an antibody currently in clinical trial[41]. However, while human anti-TIM3 antibodies pursued therapeutically are largely Fc receptor-silent[27], the rat IgG2a subclass of the anti-TIM3 antibody RMT3-23 used here does effectively engage Fcγ receptors and can thus trigger corresponding effector functions. Thus, these caveats need to be considered when translating our findings into therapeutic development.

Nevertheless, the focus on a direct effect of checkpoint blockade regimens on CTL responses is important as a complement to current inhibitory receptor blockade therapies. CTLA-4 and PD-1 blockade improve the anti-tumor immune response amongst patients with melanoma, NSCLC, and hematological cancers[13,42], however, frequently with substantial adverse immunological effects. PD-1 and CTLA-4 are expressed during T cell differentiation and by CTL throughout the body. Moreover, they are expressed by other cell types, such as myeloid cells and regulatory T cells, respectively. Action on myeloid cell types and the generation of new T cell clones at priming, potentially including autoreactive ones, maybe a principal mechanism of PD-1 and CTLA-4 blockade with only secondary effects on CTL killing[43,44]: In basal and squamous cell carcinoma patients, anti-PD-1 does not lead to the activation of existing anti-tumor T cell clones but to tumor enrichment of new ones[6]. In a mouse melanoma model deletion of PD-1 in T cells does not enhance anti-tumor immunity; however, deletion of PD-1 in myeloid cells does[5]. We showed that blocking PD-1 in the in vitro interaction between CTL and tumor cells actually impaired, rather than improved, killing and the necessary cytoskeletal polarization steps[7]. The direct role of $A_{2A}R$ and TIM3 in regulating CTL—tumor cell interaction thus generates a promising contrast to PD-1 and CTLA-4. Accordingly, $A_{2A}R$ antagonists and anti-TIM3 are already explored in early-stage clinical trials, often in combination with anti-PD-1[27,45]. Anti-TIM3 mAb which has recently completed Phase I trials include LY3321367 in advanced

relapsed/refractory solid tumors (Eli Lilly) and Sabatolimab (MGB453) in combination with anti-PD-1 in advanced solid tumors (Novartis)[46,47]. Currently, there are at least four $A_{2A}R$ antagonists which are in Phase II trials, NIR-178 (Novartis) with anti-PD-1 in multiple solid tumors and diffuse large B-cell lymphoma, PBF-509 (Novartis) in non-small cell lung carcinoma with anti-PD-1, NCT02754141 (Astra Zeneca) with anti-PD-L1 and anti-CD73 in prostate cancer and an $A_{2A}/A_{2B}$ antagonist AB928 (Arcus Biosciences) with chemotherapy in pancreatic cancer[48]. Our work may provide an incentive to include the investigation of CTL polarization as an integral part of such trials.

Despite the promise of targeting TIM3 and $A_AR$ in early clinical trials, it is still unclear whether or not blocking $A_{2A}R$ and/or TIM3 will lead to fewer autoimmune side effects. Adenosine, $A_{2A}R$ and TIM3 not only suppress T cell function but also that of myeloid cells[49,50]. However, effects of adenosine in myeloid cells may be mediated by another adenosine receptor, $A_{2B}R$[51]. $A_{2A}R$ expression in T cells also contributes to thymic development and in the maintenance of a quiescent state among naïve T cells[52]. Autoimmunity upon melanoma rejection in $A_{2A}R$-deficient mice has been observed[19]. Nevertheless, approval of the $A_{2A}R$ antagonist Istradefylline for the treatment of Parkinson's Disease[53] and several early phase clinical trials of the $A_{2A}R$ antagonist CPI-444 in cancer indicate that autoimmune side effects of $A_{2A}R$ blockade can be minimal.

CTLA-4 and PD-1 have the greatest efficacy in tumors that possess large numbers of infiltrated CD8+ TIL, which are predominantly suppressed by inhibitory receptor expression[13]. However, such 'hot' tumors represent only a minority of all cancer types, prominently melanomas. In contrast, cold tumors lack an immune infiltrate almost entirely. In altered immunosuppressed and altered-excluded tumors cancer-specific CTL priming and infiltration do occur but tumor-mediated immune suppression, rather than relying predominantly on inhibitory receptors, employs alternate suppressors such as adenosine and prostaglandin E2, Tregs, and inhibitory cytokines such as IL-10 and TGF-β[13]. Moreover, infiltration of CTL into tumors can often be partial, reaching only the peripheral stromal regions of the tumor mass. Therapeutic approaches to increase tumor infiltration by CTL are therefore important in extending immunotherapy to cold and altered phenotype cancers. CTL tumor infiltration is enhanced in $A_{2A}R$-deficient mice, mice with selective $A2_{A}aR$ deletion in T cells and upon $A_{2A}R$ agonist treatment, albeit in experiments using hot mouse tumor models[22–24]. In various mouse models, CD73 activity on non-hematopoietic cells limited tumor infiltration of T cells[54]. Increased tumor infiltration of CD8+ CTL upon $A_{2A}R$ blockade may occur because $A_{2A}R$ regulates the endothelium[55]. In addition, CTL-intrinsic mechanisms of enhanced tumor infiltration are likely. We showed that $A_{2A}R$ engagement could efficiently block CL4 CTL infiltration of spheroids (Fig. 6c). $A_{2A}R$ engagement directly suppresses CD8+ T cell migration through inhibition of the KCa3.1 ion channel[56]. Adenosine may also regulate the infiltration of other in immune cell types, as blockade of CD73 leads to increased dendritic

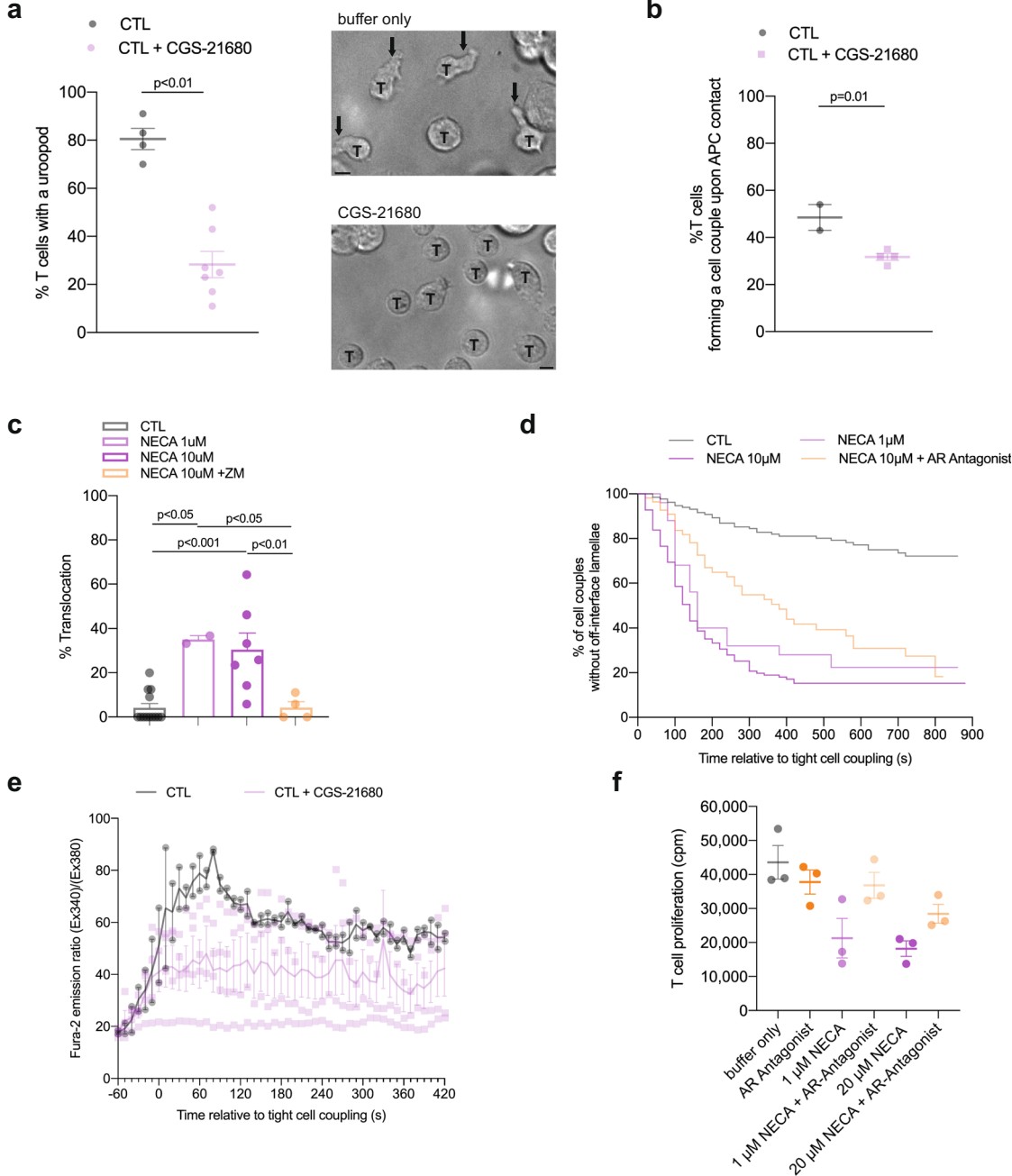

**Fig. 7 A$_{2A}$R suppresses the cytoskeletal polarization of CTL. a** Percentage of in vitro CL4 CTL with a uropod ± 1 μM CGS-21680 ± SEM. $N = 4$ independent experiments, 220/648 T cells analyzed. Representative images of CL4 T cells (labeled with 'T') in a field with Renca APC treated ± 1 μM CGS-21680. Arrows indicate uropods. Scale bar = 5 μm. **b** Frequency of CL4 T cells that form a tight cell couple upon contact with a Renca APC incubated with 2 μg/ml K$^d$HA peptide ± SEM. $N = 2$ independent experiments, 68/234 T cells analyzed. **c, d** Imaging of the interaction between in vitro CL4 CTL with K$^d$HA-pulsed Renca tumor cell targets treated with NECA ± 1.25 μM A2aR antagonist ZM 241385. **c** Percentage of cell couples with translocation (1-way ANOVA) ±SEM. $N = 28$-132 cell couples per condition over ≥2 experiments. Alternate analysis of pooled data by proportion's z-test yields $p < 0.001$ for all comparisons between NECA only-treated and control or ZM 241385-treated samples. **d** Time until the formation of first off-interface lamella (Kaplan–Meier survival analysis (Log Rank)). $P < 0.01$ all comparisons. **e** In vitro CL4 CTL interacted with RencaHA cells incubated with 2 μg/ml K$^d$HA peptide ± 1 μM CGS-21680. The ratio of Fura-2 emissions at 510 nm upon excitation at 340 nm over 380 nm is given relative to time of tight cell coupling as the mean ± SEM. $N = 2$ independent experiments, 13/42 T cells analyzed. **f** CL4 T cells were primed in vitro using anti-CD3/CD28 mAb. NECA ± 1.25 μM ZM 241385 were added at 0 h and $^3$H-thymidine for the last 8 h of cell culture. Proliferation was quantified by $^3$H-thymidine incorporation (cpm) and is given as the mean ± SEM ($N = 3$ experiments). Source data are provided in Supplementary Data 6 (all but the calcium data) and Supplementary Data 7 (calcium data).

cell infiltration in the context of radiotherapy of poorly immunogenic tumors[50].

Blockade of A$_{2A}$R led to compensatory upregulation of inhibitory receptor expression as previously noted[15,57]. Compensation

between different elements of tumor-mediated immune suppression has been described before, e.g. in the upregulation of A$_{2A}$R expression upon PD-1 blockade[58]. Curiously, blocking A$_{2A}$R during vaccination leads to diminished inhibitory receptor expression

on T cells[21]. In a model of colon cancer, $A_{2A}R$ blockade does not alter PD-1 expression on $CD8^+$ TIL but reduced it on T cells in the tumor-draining lymph nodes[21]. Effects of $A_{2A}R$ blockade on inhibitory receptor expression thus may be context dependent.

A key mechanism of $A_{2A}R$ and TIM3 in regulating CTL function is the suppression of cytoskeletal polarization in T cell migration and target cell killing. We can only speculate on underlying signaling mechanisms. In principle, two scenarios are conceivable. $A_{2A}R$ and/or TIM3 could trigger signaling pathways that directly regulate cytoskeletal dynamics. Alternatively, these receptors could modulate general proximal T cell signaling that is known to control cytoskeletal dynamics. Consistent with direct cytoskeletal regulation, $A_{2A}R$ signaling through cAMP/protein kinase A results in inhibition of RhoA and Cdc42 in leukocytes during cell-cell adhesion[59]. In cardiomyocytes, adenosine receptor agonists prevent RhoA activation and cofilin-mediated actin polymerization[60]. Alternatively, $A_{2A}R$ binds α-actinin[61]. Such binding could lead to the sequestration of α-actinin whose recruitment to the T cell/APC interface is required for effective T cell polarization[62]. Consistent with modulation of general proximal T cell signaling, $A_{2A}R$ and TIM3 could converge on the tyrosine kinase Lck. Elevated cAMP levels in response to $A_{2A}R$ engagement can enhance Csk activity[15,63], leading to inhibitory phosphorylation of Lck. When not engaged by ligand, TIM3 binds BAT3, a molecule that maintains a reservoir of Lck at the immune synapse and thus lowers the threshold for TCR signaling. TIM3 binding to Galectin-9 or CEACAM-1 releases BAT3, disabling Lck pre-localization at the immunological synapse[64]. Also consistent with TIM3-mediated regulation of general proximal T cell signaling, Y256 and Y263 of the TIM3 cytoplasmic domain are phosphorylated in response to T cell activation, can associated with the Src family kinase Fyn and the p85 subunit of phosphatidylinositol 3-kinase and lead to elevated tyrosine phosphorylation of phospholipase C γ1[65]. Phospholipase C γ1-dependent generation of diacylglycerol is a key regulator of multiple steps in cytoskeletal polarization[66]. While we have consistently related $A_{2A}R$ and TIM3 engagement to impaired cytoskeletal polarization here, future work will be required to establish the signaling mechanisms that underpin this relationship.

In summary, our work supports blocking of $A_{2A}R$ and TIM3 as an attractive complement to PD-1 and CTLA-4 blockade in anti-tumor immunotherapy. $A_{2A}R$ and TIM3 blockade directly enhanced the ability of CTL to polarize towards and kill tumor target cells in tumors and tumor spheroids and may thus reactivate tumor-resident suppressed CTL. In addition, regulation of tumor and spheroid infiltration by $A_{2A}R$ promises therapeutic potential in cold and altered immunity tumors.

## Materials and methods

**Mice.** Thy1.2$^{+/+}$ BALB/c, (Charles River, Oxford, UK) and Thy1.1$^{+/+}$ CL4 TCR transgenic mice [Research Resource Identifier (RRID): IMSR_JAX:005307] were maintained at the University of Bristol Animal Services Unit. All mouse experiments were compliant with UK Home Office Guidelines under PPL 30/3024 to DJM as reviewed by the University of Bristol AWERB (Animal welfare and ethical review body) committee.

**Antibodies.** Antibodies used are described in the order: antigen, fluorescent label, clone, supplier, dilution, RRID:

For flow cytometry:
FcBlock no azide (for blockade of Fc receptors) 2.4G2 BD Biosciences 1:50 RRID:AB_2870673
CD8α FITC 53-6.7 BD Bioscience 1:100 RRID:AB_394569
CD8β PeCy7 YTS156.7.7 BioLegend 1:200 RRID:AB_2562777
CD4 AF700 CK1.5 BioLegend 1:100 RRID:AB_493698
CD39 PerCP-Cy5.5 24DMS1 eBioscience 1:100 discontinued
CD73 BV605 TY/11.8 BioLegend 1:100 RRID:AB_2561528
TIM3 PE B8.2C12 BioLegend 1:100 RRID:AB_1626177
TIM3 BV605 RMT3-23 BioLegend 1:100 RRID:AB_2616907
TIGIT APC 1G9 BioLegend 1:100 RRID:AB_10962572

LAG3 PeCy7 C9B7W eBioscience 1:200 discontinued
PD1 BV785 29 F.1A12 BioLegend 1:200 RRID:AB_2563680
TCRβ AF647 H57-597 BioLegend 1:200 RRID:AB_493346
Thy1.1 FITC OX-7 BD Bioscience 1:100 RRID:AB_395588
Thy1.1 PerCP-Cy5.5 OX-7 BioLegend 1:100 RRID:AB_961438
CEACAM1 APC CC1 BioLegend 1:100 RRID:AB_2632612
For blocking and T cell priming
TIM3 no azide (for in vitro/in vivo blockade) RMT3-23 BioXcell In Vivo mAb in vivo: 100 μg/mouse in vitro: 10 μg/ml RRID:AB_10949464
Isotype control for anti-TIM3 Rat IgG2a 2A3 no azide (for in vivo/in vitro blockade) BioXcell In Vivo mAb in vivo: 100 μg/mouse in vitro: 10 μg/ml RRID:AB_1107769
CD8β no azide (for in vivo depletion) 53-5.8 BioXcell InVivoMAb 100 μg/mouse RRID:AB_2687706
Thy1.1 no azide (for in vivo depletion) 19E12 BioXcell InVivoMAb 250 μg/mouse RRID:AB_2687700
CD3ε no azide (for in vitro priming) 145-2C11 BioXcell InVivoMAb 10 μg/ml RRID:AB_1107634
CD28 no azide (for in vitro priming) 37.51 BioXcell InVivoMAb 1 μg/ml RRID:AB_1107624
For immunohistochemistry:
FcBlock no azide (for blockade of Fc receptors) 2.4G2 BD Biosciences 1:50 RRID:AB_2870673
CD8α no azide 53-6.7 BioLegend 1:500 RRID:AB_312741
Rabbit H + L AF488 Life Technologies 1:1000 RRID:AB_143165
Rabbit H + L AF405 Life Technologies 1:1000 RRID:AB_221605
Rat IgG2a,κ BioLegend 1:500 RRID:AB_326523
Rat IgG H + L AF594 ThermoFisher 1:2000 RRID:AB_141374
FOXP3 no azide FJK-16S ThermoFisher 1:100 RRID:AB_467575
FOXP3 APC FJK-16S ThermoFisher 1:40 RRID:AB_469457
Thy1.1 FITC OX-7 BD Bioscience 1:100 RRID:AB_395588
Isotype control for Thy1.1 Mouse IgG1,κ FITC BD Bioscience 1:100 RRID:AB_395505
CD19 (dump) BV510 6D5 BioLegend 1:100 RRID:AB_2562136
TCRβ SB645 H57-597 ThermoFisher 1:200 RRID:AB_2723704
CD4 PE-Cy5.5 RM4-5 ThermoFisher 1:3000 RRID:AB_1121830
CD8β PE-Cy5 H35-17.2 ThermoFisher 1:3000 RRID:AB_657770
CD25 VioBright-FITC 7D4 Miltenyi 1:200 RRID:AB_2784091
CD73 BV605 TY/11.8 BioLegend 1:100 RRID:AB_2561528
CD39 PerCP-eFluor710 24DMS1 ThermoFisher 1:100 RRID:AB_10717953

**Cell culture.** Murine Renal Carcinoma cell line (RRID:CVCL_2174) and Phoenix retrovirus-producing cell line (RRID:CVCL_H717) were maintained as previously described[7].

To generate in vitro CL4 CTL, CL4 mouse spleens from 6-12-week-old male or female mice were macerated. Red blood cells were lysed using ACK Lysis buffer (Gibco, ThermoFisher, Gloucester UK), and the remaining splenocytes were resuspended in complete medium, RPMI-1640 with 10% FBS and 50 μM 2-mercaptoethanol. $5 \times 10^6$ cells were seeded into each flat bottomed 24 well tissue culture plate with 1 μg/ml of $K^d$HA peptide (IYSTVASSL$_{[518-526]}$) from influenza virus A/PR/8/H1N1, for 24 h at 37 °C. After 24 h, cells were washed five times in RPMI (Gibco) and reseeded into 24 well plates at $5 \times 10^6$ cells per well in 2 ml complete medium containing 50 units/ml of recombinant human IL-2 (National Institutes of Health/NCI BRB Preclinical Repository). Retroviral transduction was performed if required as previously described[7]. CL4 T cells were then passaged every 12–24 h using fresh IL-2 containing complete medium. Where indicated, 1 or 10 μM NECA (Sigma), 1.25 μM ZM 241385 (Santa Cruz), or 1 μM CGS-21680 (Tocris) were added to cell culture. DMSO was used as vehicle control. ZM 241385 properties, including selectivity for adenosine receptors, are detailed at https://www.abcam.com/zm-241385-a2a-antagonist-ab120218.html.

For $^3$H-thymidine proliferation assays[67], Clone 4 T cells were primed using anti-CD3/CD28 mAb stimulation or with mature agonist-peptide pulsed dendritic cells in a flat bottomed 96 well plate and cultured for the desired time at 37 °C. $^3$H-thymidine (Amersham Life Science, London, UK) was added for the final 8 h of cell culture at 1.45 mBq/ml. To harvest, the entire plate was frozen at −20 °C for at least 24 h and subsequently defrosted to produce cell lysis. $^3$H-thymidine incorporation was measured using a 96 well Tomtec harvester and a Microbeta scintillation counter (PerkinElmer).

Extraction of T cells from tumor tissue was carried out using magnetic-activated cell sorting (Miltenyi) and flow cytometry[7].

**Tumor growth and treatment experiments.** Six-week-old Thy1.2$^{+/+}$ BALB/c mice were injected subcutaneously, in the dorsal neck region, with $1 \times 10^6$ RencaHA tumor cells in 100 μl PBS. Tumor measurements and treatment commenced at day 12, when tumors of ~$5 \times 5$ mm diameter were palpable. For adoptive transfer experiments, tumor-bearing mice were injected i.v. at day 12, with $5 \times 10^6$ day 5 CL4 CTL (see above). For in vivo immunotherapy, control mice received 100 μl vehicle (15% vol/vol DMSO, 15% vol/vol Cremophore EL, 70% vol/vol PBS) ± 100 μg/mouse isotype control (Rat IgG2a, 2A3, BioXcell InVivoMAb)[68]. Treated mice received combinations of 10 mg/kg ZM 241385 injected

intraperitoneally in 100 μl vehicle and 100 μg/mouse anti-TIM3 mAb (RMT3-23, BioXcell InVivoMAb) injected intraperitoneally in 100 μl PBS on alternate days throughout the experiment. Thus, ZM 241385-treated mice did not receive the mouse isotype control antibody. However, earlier work has shown that mouse isotype control antibody does not affect Renca tumor growth in vivo[7]. Tumors were measured on alternate days using calipers and the volume was calculated using the modified elliptical formula: Volume = 0.5 × length × width[2]. CL4 CTL restimulation in vivo was achieved by i.p. injection of 200 μl of PBS containing 1200 HA units of influenza virus A/PR/8/H1N1 virus, as in[30]. Rechallenge with tumor cells was achieved following stable remission (remained < 5 mm diameter for 8 days) by injecting a further $1 \times 10^6$ RencaHA cells subcutaneously into the dorsal neck region in PBS. Depletion of CD8[+] or Thy1.1[+] T cells was performed by injection of depleting mAb (anti-CD8, 53-5.8, BioXcell InVivoMAb, 100 μg/mouse, Thy1.1 19E12 BioXcell InVivoMAb, 250 μg/mouse).

**Imaging and image analysis**. For live-cell imaging of immune synapse formation and CL4 T cell morphology $1 \times 10^6$ Renca tumor target cells were pulsed with 2 μg/ml K[d]HA for 1 h at 37 °C. Cells were then resuspended at $1 \times 10^6/400$ μl Imaging Buffer (PBS, 10% FBS, 1 mM CaCl₂, 0.5 mM MgCl₂). To image the increase in the cytoplasmic Ca²⁺ concentration, CL4 CTL were incubated with 2 μM Fura-2 AM (Molecular Probes) for 30 min at room temperature in imaging buffer and washed twice thereafter.

40,000 Clone 4 CTL or TIL in 5 μl imaging buffer were plated with 1–1.5 μl Renca target cells (preceding paragraph) in 50 μl imaging buffer, in a 384-well, glass-bottomed imaging plate (Brooks). If reagents such as NECA were included in cell culture, they were also added to imaging buffer at an equivalent final assay concentration. Every 10 s for 15 min, one bright-field differential interference contrast (DIC) image, one fluorescence image with excitation at 340 nm, and one fluorescence image with excitation at 380 nm were acquired at 37 °C with a 40x oil objective (NA = 1.25) on a Leica DM IRBE-based wide-field system equipped with Sutter DG5 illumination and a Photometrics Coolsnap HQ2 camera.

Using MetaMorph (Molecular Devices) for analysis of DIC images, tight cell couple formation was defined as the first time point at which a maximally spread immune synapse formed, or two frames after initial cell contact, whichever occurred first. Prior to cell coupling, a T cell was deemed to have a uropod when it displayed a membrane extension that is opposite the leading edge with a region of inverse curvature relative to the entire cell at its base for a duration of at least 1 min. To assess CTL and TIL morphology in cell couples with tumor target cells, every DIC frame after tight cell couple formation was assessed for the presence of off-synapse lamellae, defined as transient membrane protrusions pointing away from the immune synapse, followed by retraction. To determine CTL translocation over the Renca cell surface, the position of the immune synapse on the RencaHA target cell was compared to the position at cell coupling. If the T cell had migrated by a distance greater than the diameter of the immune synapse, this was classed as translocation. For calcium analysis, field-averaged background fluorescence was subtracted from the 340 nm and 380 nm excitation fluorescence data, and the ratio of the Fura-2 images upon excitation at 340 versus 380 nm was determined in a circular region of interest of the dimensions of the T cell.

For Microscope-based Cytotoxicity Assays, the IncuCyte™ Live Cell analysis system and IncuCyte™ ZOOM software (Essen Bioscience) were used to quantify target cell death. $1 \times 10^6$ Renca cells transfected to express the fluorescent protein mCherry were either untreated (control) or pulsed with 2 μg/ml K[d]HA peptide for 1 h. Cells were centrifuged and resuspended in a 3.33 ml Fluorobrite complete medium (ThermoFisher) to a concentration of 15,000 cells/50 μl. Cells were plated in each well of a 384 well PerkinElmer plastic-bottomed view plate and incubated for 4 h to adhere. CL4 T cells were FACS sorted, and 15,000 cells were added to the plate in 50 μl Fluorobrite medium 4 h after target cells were plated, yielding a 1:1 effector to target ratio. Images were taken every 15 min for 14 h at 1600ms exposure using a 10x lens. The total red object (mCherry target cell) area (μm²/well) was quantified at each time point. The T cell killing rate was determined as the linear gradient of the red object data at its steepest part between the time at which Control Clone 4 CTL started killing until they had eradicated the Renca cell monolayer. The T cell killing rate was normalized to the growth of Renca (control) cells which were not pulsed with cognate HA antigen.

**Spheroids**. RencaHA tdTomato cells were resuspended at a concentration of $1 \times 10^5$ cells/ml, mixed with Matrigel (Corning) at 4 °C, seeded in a 24-well plate at a final concentration of 500 cells per Matrigel dome, and left to solidify for 10 min at 37 °C. 2 ml cell medium was added to each well and cells incubated at 37 °C for 11 days. Each Matrigel dome was washed twice in PBS and incubated for 30 min with 1 ml of Cell Recovery Solution (Corning). Spheroids were collected in a 15-ml Falcon tube and pulsed with K[d]HA peptide at a final concentration of 2 μg/ml for 1 h. Pulsed spheroids were re-embedded in Matrigel together with $5 \times 10^5$ primed CL4 CTL per Matrigel dome. Matrigel domes were dissolved for analysis of spheroid-infiltrating T cells after 16 h: Spheroids were washed twice in PBS and incubated with 1 ml of Cell Recovery Solution (Corning). Spheroids were collected, washed through a 70 μm sieve, and then disaggregated to retrieve T cells in 500 μl of imaging buffer for immediate FACS sorting.

**Spheroid imaging**. Spheroids were grown as described in the preceding paragraph. On day 10, CL4 CTL that had been retrovirally transduced to express the GFP-tagged protein of interest (TIM3-GFP or F-tractin-GFP) were sorted by flow cytometry and incubated in IL-2 medium for 1 h ± 10 μg/ml anti-TIM3 mAb (Clone RMT3-23) or 1 μM CGS-21680 where appropriate. Meanwhile, spheroids were dissociated from Matrigel and resuspended into fresh Matrigel at a concentration of ~8 spheroids/μl. 50 μl of the spheroid-Matrigel suspension was separated into Eppendorf tubes for each treatment group, followed by the addition of 200,000 sorted T-cells per tube. 50 μl of Matrigel, containing spheroids and T cells, was plated into each well of a 24-well tissue culture plate. After Matrigel had set, 1 ml of Fluorobrite medium was added to each well, containing 1.5 μM DRAQ7 viability dye ± 10 μg/ml anti-TIM3 mAb or 1 μM CGS-21680. Images were acquired every 2 h post-plating CTL with spheroids in 3 μm z steps from the bottom of the spheroid to its widest point, usually 40 steps, for 12 h using a Leica SP8 AOBS confocal microscope with a 10x HC PL Fluotar lens (NA = 0.3). To obtain measurements of SIL density and spheroid dead volumes, raw data were pre-processed and semi-automatically analyzed using a custom-written Cancer Segmentation workflow for the Fiji[69] plugin, MIA (v0.9.26) and its MIA_MATLAB (v1.1.1) package, available at GitHub via Zenodo: https://doi.org/10.5281/zenodo.2656513 and https://doi.org/10.5281/zenodo.4769615, respectively. The corresponding.mia workflow files are available at https://doi.org/10.5281/zenodo.5511888. Briefly, the imaged stacks were mirrored and concatenated along the z-axis to produce pseudo-complete spheroids. These spheroids were binarised and segmented using connected-components labeling[70]. To account for fragmented spheroid segmentation arising from gaps in labeling, in particular towards spheroid centers, spheroids were fit with alpha shapes[71] using the MATLAB implementation. Adjacent spheroids which had become merged during processing were separated with a distance-based watershed transform[70]. T-cells and dead volumes were individually segmented from their respective fluorescence channels using similar threshold and labeling-based processes, albeit without the alpha shape step.

**Flow cytometry staining with its principal component analysis**. MACS-purified TIL was resuspended in PBS at a concentration of $1 \times 10^6$ cells/ml. $2.5 \times 10^5 – 1 \times 10^6$ cells for each condition resuspended in 100 μl PBS per tube with 1 μl/100 μl Zombie Aqua Fixable Live Cell Detection reagent (BioLegend). Tubes were incubated for 15 min in the dark at room temperature. Cells were washed in 3 ml FACS buffer and resuspended in 100 μl per tube FcBlock (BD Biosciences) for 15 min at 4 °C. Cells were washed in 3 ml FACS buffer, pelleted, and resuspended in 100 μl FACS buffer per tube with antibody at the required concentration (antibody section above) and then incubated for 30 min at 4 °C. Antibody concentration was determined by titration using five concentrations centered around the manufacturer's recommended protocol. Cells were washed in 3 ml FACS buffer before being fixed in 1% paraformaldehyde and analyzed within 5 days using a Fortessa Flow Cytometer and BD FACSDiva Software (BD Biosciences). Flow cytometric data were analyzed using FlowJo™ (Treestar) software. Gating was performed using fluorescence minus one (FMO) samples for each antibody stain. In the principal component analysis of inhibitory receptor expression upon mouse treatment with ZM 241385 all combinations of % positive were used as input data as listed in the source file for Fig. 2. A custom script to execute the analysis is available at GitHub: https://github.com/ge8793/rencaPCA.

**Immunohistochemistry**. Tumors were harvested and cut in half. Half of the tumor was used for CTL function assays. The other half was placed in 2.35 ml RPMI on ice. Within 1 h of harvest, tumors were snap-frozen in OCT compound (Tissue Tek) on a square of cork in an isopentane bath, hovering above Liquid Nitrogen. Tumors were sectioned into 5 μm sections and mounted on slides. On day one of staining, acetone was cooled to −20 °C. Slides were allowed to air dry for 10–20 min before being fixed in acetone for 10 min on ice. Slides were dried again then washed three times in PBS. Slides were dried in the area around the section and a border marked using a hydrophobic pen (ImmEdge). Sections were blocked with 2.5% horse serum (Vector) for 30 min then washed three times in PBS. Sections were incubated with primary antibodies or isotype controls in 1% BSA/PBS (Sigma-Aldrich) overnight at 4 °C or room temperature for 1 h (antibody section above). On day 2 of staining, slides were washed three times with PBS and incubated with secondary antibody prepared in 1% BSA/PBS for 30–60 min at room temperature (antibody section above). Slides were washed three times in PBS, with the second wash being performed in a shaker for 10 min. Hoechst stain (ThermoFisher) was applied for 10 min, followed by three PBS washes. Slides were fixed in 1% PFA for 10 min then washed twice in PBS, once in Glycine (0.3 M) (Fisher Chemical) for 10 min, and one final wash in PBS. Coverslips were mounted in prolong gold antifade reagent and slides were left to cure at room temperature for 24 h and images were acquired using a mark and find experiment on the same confocal microscope as used for spheroid imaging. Images were analyzed using ImageJ (Fiji).

**Statistics and reproducibility**. The Power of in vivo experiments was designed to reach >80%. Experimental group size was determined using the equation: $n = (2/$

(standardized difference2)) x cp,power, where $n$ = sample size per group determined using the formula, $d$ = standardized difference = measurable difference in tumor volume / standard deviation, cp,power = constant for $p < 0.01$ and power at 80% defined using standard Altman's Nomogram = 11.7.

Samples were compared using independent sample $t$-tests for two-sample comparisons. To determine the effect of one or more independent variables on one dependent variable across >2 groups, One-way and Two-way ANOVA were used. Where proportions were compared, Fisher's exact Boschloo or a proportion's $z$-test was used. SPSS statistics and Prism were used to execute analyses.

In analyses of in vivo tumor growth and survival, the following factors were considered:

1. Experiments using Single Agent treatment of RencaHA tumors: when single immunotherapy was administered, tumor growth was unidirectional and tumors did not shrink; therefore, repeated measures ANOVA was utilized to compare growth curves. Kaplan–Meier analysis was used to compare survival (Mantel Cox, Log rank), since there were only two outcomes in the study, survival or death. Mice were censored if tumors were < maximal allowable tumor size (MATS) or if culled for reasons other than tumor size (such as ulceration). Mice bearing tumors > MATS were culled and recorded as dead.

2. Experiments using combined treatment with Adoptive T cell transfer, TIM3 blockade, and A2aR blockade Cohorts of 14 mice per group were injected to achieve a minimum 4–6 tumors per group of uniform size (3–5 mm diameter) between 12–14 days after injection, for each experimental replicate. Tumor growth was compared using $R$-values. To account for growth over the whole triphasic growth curve, final volume/initial volume is used because an exponential model cannot be applied to tumor growth > 1 week[72–74]. For early-stage tumors, an exponential model was fitted. $R$-value (units of $R = 1/time$) was calculated using: $R = c(dc/dt)$ where: qg = $2.76 \times 10^5$ Renca tumor cell = $17–25\,\mu M$ diameter in vitro = $3.61\,nm^3$ volume using modified elliptical formula. Therefore $2.76 \times 10^5$ Renca cells make up $1\,mm^3$ of tumor. Each Renca cell divides three times every 24 h, dt = 8 h, dc = difference in cell number = (qg × final volume in $mm^3$) – (qg × starting volume in $mm^3$), $c$ = number of cells at start = (qg × starting volume in $mm^3$)[72–74]. Progression-free survival was calculated using Cox's regression with covariates (SPSS). Hazard ratio of relapse was calculated using SPSS statistics.

Analysis of flow cytometric data: flow cytometric data were analyzed using FlowJo™ (Treestar). Gating was performed using Fluorescence Minus One control samples. Boolean gating tool was used to determine all possible combinations of expression of certain markers. Percentage expression of co-inhibitory receptors as determined by flow cytometry were arcsine square root transformed prior to analysis. Tumor volume was $log_{10}$ transformed. Principal component analysis (PCA) was performed using RStudio™ to analyze the combination expression of six markers as determined by flow cytometry, and tumor volume. Tumors with a growth rate >1 standard deviation from the mean were excluded from comparisons. After transformation, each value $(x)$ within the expression data and volume data were standardized $(x^*)$ to give a mean $(m)$ of zero and a standard deviation (SD) of one using the formula $x^* = (x - m)/sd$. PCA was then performed using RStudio and the packages 'FactoMineR' and 'Factoextra'. Principal components (PC) 1–5 were selected for inclusion based on eigenvalues >1. Cos2 values were used to confirm the quality of representation of each variable within 2-dimensional factor maps and were calculated as the square of the variable's co-ordinates. The contribution of variables to PCs 1–4 was calculated using the formula (var.cos2 * 100) / (total cos2 PC) to produce $P$-values.

**Reporting summary**. Further information on research design is available in the Nature Research Reporting Summary linked to this article.

## Data availability
Two large data sets, the flow cytometry data underpinning the principal component analysis to determine inhibitory receptor expression upon A2aR blockade in Fig. 2 and the spheroid imaging data in Fig. 6 are accessible through an open data repository of the University of Bristol at https://doi.org/10.5523/bris.11ocsor59owa32ihsxmf0qzj3s. All other source data are provided as Supplementary Data 1 through 7 and corresponding raw imaging data will be made available upon request.

## Code availability
A custom-written script to execute the principal component analysis of inhibitory receptor expression is available at GitHub: https://github.com/ge8793/rencaPCA.
A custom-written Cancer Segmentation workflow for spheroid analysis for the Fiji plugin, MIA (v0.9.26) and its MIA_MATLAB (v1.1.1) package, is available at GitHub via Zenodo: https://doi.org/10.5281/zenodo.2656513 and https://doi.org/10.5281/zenodo.4769615, respectively. The corresponding.mia workflow files are available at https://doi.org/10.5281/zenodo.5511888.

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

## Acknowledgements

We acknowledge the University of Bristol Flow Cytometry Facility with Andrew Herman and Lorena Sueiro-Ballesteros and the Wolfson BioImaging Facility with Katie Jepson for experimental support, Alan Hedges for statistical advice, and Mick Bailey for contributions to the principal component analysis. This work was supported by grants from the Wellcome Trust via the University of Bristol Elizabeth Blackwell Institute (Wellcome Trust ISSF2 grant 105612/Z/14/Z to G.L.E.), the Wellcome Trust (201254/Z/16/Z/ to G.L.E., 102387/Z/13/Z/ to R.A.) and the University of Bristol Cancer Research Fund (to C.W.). G.L.E was also supported as an associate of the Wellcome Trust GW4-CAT scheme. C.C.W.W. was supported by the MRC GW4 DTP. E.J.M. was supported by a Wellcome Trust GW4-CAT Fellowship. For the purpose of Open Access, the author has applied a CC BY public copyright licence to any Author Accepted Manuscript version arising from this submission.

## Author contributions

G.L.E. designed and carried out experiments, analyzed data, and wrote the paper. C.C.W.W. designed and carried out experiments and analyzed data. R.A. designed and carried out experiments and analyzed data. E.J.M. designed and carried out experiments and analyzed data. H.A. designed and carried out experiments and analyzed data. S.J.C. designed methods for image analysis, G.G. carried out experiments and analyzed data. C.W. designed and carried out experiments, analyzed data, and wrote the paper. D.J.M. designed and carried out experiments, analyzed data, and wrote the paper.

## Competing interests

The authors declare no competing interests.
