## [Transparent Peer Review File · Communications Biology]

Reviewers' comments:

Reviewer #1 (Remarks to the Author):

The manuscript by Edmunds et al. examines the relationship between the adenosine receptor A2Ar and the checkpoint molecule Tim-3. Most experiments use Renca tumors expressing the surrogate antigen HA, along with CL4 TCR Tg T cells, which recognize HA presented by H-2 Kd. A few experiments also make use of an in vitro tumor spheroid model. Thus, the authors found that Tim-3 is further upregulated on tumor-infiltrating CD8+ T cells after inhibition of A2Ar, suggesting that this represents a compensatory pathway. Consistent with such a hypothesis, a widely used Tim-3 mAb synergizes with A2Ar inhibition to slow tumor growth and enhance anti-tumor immunity. The authors then go on to examine the effects of A2Ar and/or Tim-3 blockade on events known to be dependent on cytoskeletal polarization, including polarization of T cells toward target cells, cytotoxicity and tumor infiltration.

The data that are presented here appear to be properly controlled and robust. I do, however, have some concerns regarding the larger conclusions that the authors draw from their findings, as well as a few suggestions for improving data presentation.

1. A general issue is that there is a dearth of example flow cytometry and imaging data to accompany the graphical data shown in most figures. For example, the PCA plot in Fig. 1E is useful, but it would be helpful to see staining of Tim-3, etc. not just in the control situation, but also after A2Ar antagonist treatment. Similarly, in Fig. 5, the TCR polarization away from the synapse is only reported in graphical form, without any example imaging. Finally, the polarization data in Fig. 7 would benefit from the inclusion of some example figures and gates or masks showing how the measurements were made. I know these things are described to some extent in the Methods, but visual examples would be helpful for assessing the conclusions.

2. The authors do present several lines of evidence which suggest that the effects of Tim-3 on the cytoskeleton are a primary mechanism by which Tim-3 acts to limit anti-tumor immunity. My interpretation of their data is that this model is still preliminary and is more correlative at this point. The authors should soften their conclusion in this regard or provide more direct data. Thus, we know that TCR signaling itself does modulate the cytoskeleton during T cell activation, as do other receptors like integrins, which themselves are affected by cytoskeleton changes. How do the authors envision this circuit working at a biochemical level?

3. I also think the assertion that CD8+ T cells are the sole driver of the effects of Tim-3 is overstated. As the authors confirm here, Treg are key cells for the generation of adenosine in the tumor microenvironment, through expression of CD39 and CD73. However, the authors do not discuss, or address experimentally, the fact that TIL Treg greatly upregulate expression of Tim-3, as shown by multiple groups in different tumor types. Thus, some of the effects of Tim-3 mAb reported here could be due to modulation of Treg activity or numbers. The more reductionist experiments in the spheroids do not rule this out, as they rely on use of an A2Ar agonist. In addition, depletion of CD8+ T cells is a rather broad intervention, so it was not very surprising that this intervention reversed the modulation of tumor growth. Indeed, even positive effects of Treg depletion or inhibition require the presence of CD8+ T cells in the tumor.

Reviewer #2 (Remarks to the Author):

In this manuscript, Edmunds et al. investigated the role of A2A adenosine receptor and TIM3 in inhibiting the killing of tumor cells by cytotoxic T lymphocytes and suggested that the mechanism behind could be cytoskeletal polarization. The results are novel and interesting. I have some minor comments for the authors to considerate.

P6, L165-6: "To determine whether A2aR suppresses anti-tumor immunity in the RencaHA model, we treated RencaHA tumor-bearing mice with the A2aR antagonist ZM 241385 (Fig. 1)"

I finally found a ZM241385 concentration in the legend of figure 2 (1.25 μ M). Did the authors use the same concentration in all other experiments? Also, it should be noted that at this concentration (1.25 μ M), ZM241385 is an antagonist for both A2A and A2B (J Pharmacol Exp Ther. 2007 Feb;320(2):637-45. doi: 10.1124/jpet.106.111203. Epub 2006 Oct 31.), which may affect the interpretation of results. Do CD8 T cells express only A2A not A2B adenosine receptors? The authors may consider repeating some key results with a lower concentration of some relatively more selective A2A antagonists such as SCH442416.

In some experiments, the A2A agonist CGS21680 is used, while the nonselective agonist NECA is used in some other experiments. Again, it should be noted that NECA and ZM241385 are A2B agonist and antagonist, respectively. Although the potential role of A2B in myeloid cells is mentioned (ref. 38), it seems that role of A2B may not be limited to those cells, which could be discussed in this manuscript.

Fig. 6, the concentration of CGS21680 used is 1 μ M, is it the same concentration used in other experiments?

The authors could cite the ref. "J Biol Chem. 2003 Sep 26;278(39):37545-52. doi: 10.1074/jbc.M302809200. Epub 2003 Jul 1."

Reviewer #3 (Remarks to the Author):

Edmunds et al., demonstrate the immunosuppressive functions of Adenosine and one of its receptors, A2aR, in a syngeneic murine renal carcinoma model and that blocking A2aR with an antagonist (ZM 241385) leads to upregulation of TIM3 on TILs. Moreover, they provide novel insights into the co-operative functions of A2aR and TIM3 in regulating the cytoskeletal polarisation and cytotoxicity of CTLs and/or their ability to infiltrate the tumour core; thereby, providing a rational for combinatorial therapies targeting these tumour-intrinsic and extrinsic pathways along with other checkpoint inhibitors, such as PD-1.

Overall, this is a well-designed study; and the data have important implications in immunotherapy and are complementary to the current published literature.

Minor comments and suggestions are, as follows:

1. Line 72-81: Consider citing a recent article by Stone et al., demonstrating Treg depletion by non-blocking CTLA-4 mAbs as a dominant mechanism of CTLA-4 checkpoint blockade, <https://www.biorxiv.org/content/10.1101/2021.07.12.452090v1>

2. Line 67: Related to above, when referring to mAb immunotherapy, it is imperative that the difference in the isotype of mAb and their interactions with Fc γ receptors (Fc γ R) are taken into account, as they may induce very different effects depending on the context, eg. <https://www.sciencedirect.com/science/article/pii/S1535610815002950>

3. Fig 2: In addition to PCA data presented in Fig 2E, Fig 2D should show representative raw FACS plots of co-inhibitory marker expression data for both ZM 241385-treated and ctrl mice.

4. Majority of the numbers are not visible in Fig 2E.

5. Line 268 & 357: correct 'didn't' to 'did not'. Same for 'doesn't' (line 575).

6. Discussion section: Given the potential translational nature of this study, it would be helpful if the authors could comment on the similarities/differences between murine and human receptors described herein. Importantly, would engagement/blockade of A2aR and TIM3 induce similar effects in a human culture setting?

7. Related to above, it would be important to comment on the current status of anti-TIM3 (eg, <https://www.nature.com/articles/s41577-019-0224-6>) and anti-A2aR clinical trials for human cancers and how this study may inform the design and development of future therapeutics targeting such pathways.

8. FACS plots should be better formatted (eg, deletion of axis and gate labels) and the axis may be adjusted for a better representation of the indicated cell populations.

9. General comment for Figures: Control group should be labelled and plotted first and more prominent and distinct colours and/or distinct lines (eg, solid vs. dotted or thicker lines) for the respective groups should be used throughout, eg, in Fig 5B, the colours for TILs, CTLs and the combination group are similar and the solid lines are narrow and in Fig 5B the symbol colours are very similar.

10. In an ideal situation, deglycosylated TIM3 and PD-1 mAbs that lack any binding to FcγRs should be used for blocking experiments to prevent additional levels of complexity when interpreting the data (see above and the following examples:

<https://www.sciencedirect.com/science/article/pii/S1535610816303920>).

Furthermore, the ZM 241385 treatment group should have ideally been treated with an irrelevant isotype control mAb (ratIgG2a) for completeness, when compared to 'ZM 241385 + anti-Tim3' treatment group.

11. The statistical analyses seem appropriate and adequate experimental details have been provided.

Christoph Wülfing
David J. Morgan
School of Cellular and Molecular Medicine
University of Bristol
Bristol, BS8 1TD
United Kingdom

Bristol, 9th September 2021

We are herewith submitting a revised version of our manuscript COMMSBIO-21-1492-T entitled 'The adenosine 2a receptor and TIM3 inhibit killing of tumor cells by cytotoxic T lymphocytes through interference with cytoskeletal polarization'. We appreciate the support for the importance and execution of our study and would like to thank the reviewers for the helpful comments. We have addressed the comments as follows.

Reviewer 1

A general issue is that there is a dearth of example flow cytometry and imaging data to accompany the graphical data shown in most figures. For example, the PCA plot in Fig. 1E is useful, but it would be helpful to see staining of Tim-3, etc. not just in the control situation, but also after A2Ar antagonist treatment. Similarly, in Fig. 5, the TCR polarization away from the synapse is only reported in graphical form, without any example imaging. Finally, the polarization data in Fig. 7 would benefit from the inclusion of some example figures and gates or masks showing how the measurements were made. I know these things are described to some extent in the Methods, but visual examples would be helpful for assessing the conclusions.

We have expanded the representative FACS data to include data of T cells from ZM241385-treated mice in Fig. 2. In addition, the entire FACS data underlying the principal component analysis in Fig. 2D are now openly accessible as described in the data availability statement. We have replaced the single images in Fig. 5A with time lapse data, as a selection of annotated time points in the figure and as the entire time lapse in supplementary videos. We have added imaging data to Fig. 7 to illustrate our uropod analysis and expanded the methods section accordingly. In addition, all spheroid imaging data underpinning Fig. 6 are now openly accessible as described in the data availability statement.

The authors do present several lines of evidence which suggest that the effects of Tim-3 on the cytoskeleton are a primary mechanism by which Tim-3 acts to limit anti-tumor immunity. My interpretation of their data is that this model is still preliminary and is more correlative at this point. The authors should soften their conclusion in this regard or provide more direct data. Thus, we know that TCR signaling itself does modulate the cytoskeleton during T cell activation, as do other receptors like integrins, which themselves are affected by cytoskeleton changes. How do the authors envision this circuit working at a biochemical level?

We agree with the reviewer that our data are correlative, albeit consistently so across a number of experiments. Following the reviewer's suggestion, we have tempered statements about the relationship between cytoskeletal and anti-tumor effects of TIM3 and A2AR in the abstract, the end of the introduction, the start of the last results paragraph and in a new first paragraph of the discussion that addresses important caveats of our study (see below). We

have also substantially expended the paragraph of the discussion dealing with potential mechanisms employed by TIM3 in cytoskeletal regulation and now discuss the binding of Fyn and p85 PI3K to phosphorylated tyrosine residues in the TIM3 cytoplasmic domain which leads to the activation of PLC γ .

I also think the assertion that CD8+ T cells are the sole driver of the effects of Tim-3 is overstated. As the authors confirm here, Treg are key cells for the generation of adenosine in the tumor microenvironment, through expression of CD39 and CD73. However, the authors do not discuss, or address experimentally, the fact that TIL Treg greatly upregulate expression of Tim-3, as shown by multiple groups in different tumor types. Thus, some of the effects of Tim-3 mAb reported here could be due to modulation of Treg activity or numbers. The more reductionist experiments in the spheroids do not rule this out, as they rely on use of an A2Ar agonist. In addition, depletion of CD8+ T cells is a rather broad intervention, so it was not very surprising that this intervention reversed the modulation of tumor growth. Indeed, even positive effects of Treg depletion or inhibition require the presence of CD8+ T cells in the tumor.

We thank the reviewer for this important caveat and now address it as part of the first paragraph of the discussion. This first paragraph that also addresses additional concerns covered later in this letter reads:

‘To build a diverse tool kit of reagents for comprehensive cancer immunotherapy, it is vital we understand mechanisms of action of mediators of immune suppression. Using a matched *in vivo*, *ex vivo* and *in vitro* spheroid approach, we have established that A2aR and TIM3 directly suppress the ability of CTL to kill tumor target cells. However, such establishment of a direct effect does not rule out the existence of additional indirect effects by which A2aR and TIM3 blockade could enhance tumor immunity. For example, TIM3 is expressed on regulatory T cells in the tumor microenvironment across many tumor models, including Renca (37). Blocking TIM3 diminishes the suppressive function of such regulatory T cells and thus promotes anti-tumor immunity (37). Our data consistently link defects in CTL cytoskeletal polarization to diminished killing. However, while diminished killing is a plausible explanation for the *in vivo* effects of TIM3 and A2aR, only by direct *in vivo* manipulation of cytoskeletal polarization, in future experiments, one can prove that such impaired polarization limits tumor immunity. The engagement of a TCR by its physiological cognate MHC/peptide ligand is critical for the investigation of cytoskeletal regulation (38). To allow such TCR engagement, the experiments here were conducted in a murine system. Previous work on adenosine and TIM3 suggests that findings are likely to be directly applicable in humans. For example, the pre-clinical efficacy of A2aR antagonists which are now licensed for control of Parkinson’s disease was initially established in mice (39, 40). Elevated expression of TIM3 on T_H1 cells is observed in both multiple sclerosis and its established mouse model of experimental autoimmune encephalitis and is required for the suppression of autoimmunity (41). Moreover, the blocking anti-TIM3 antibody used here targets the same phosphatidylserine-binding site of TIM3 as an antibody currently in clinical trial (41). However, while human anti-TIM3 antibodies pursued therapeutically are largely Fc receptor-silent (27), the rat IgG2a subclass of the anti-TIM3 antibody RMT3-23 used here does effectively engage Fc γ receptors and can thus trigger corresponding effector functions. Thus, these caveats need to be considered when translating our findings into therapeutic development.’

Reviewer 2

P6, L165-6: “To determine whether A2aR suppresses anti-tumor immunity in the RencaHA model, we treated RencaHA tumor-bearing mice with the A2aR antagonist ZM 241385 (Fig.

1)”

I finally found a ZM241385 concentration in the legend of figure 2 (1.25 μ M). Did the authors use the same concentration in all other experiments? Also, it should be noted that at this concentration (1.25 μ M), ZM241385 is an antagonist for both A2A and A2B (J Pharmacol Exp Ther. 2007 Feb;320(2):637-45. doi: 10.1124/jpet.106.111203. Epub 2006 Oct 31.), which may affect the interpretation of results. Do CD8 T cells express only A2A not A2B adenosine receptors? The authors may consider repeating some key results with a lower concentration of some relatively more selective A2A antagonists such as SCH442416.

We thank the reviewer for this important consideration. We have now added a section to the second paragraph of the results (see below) in which we use the pharmacokinetics following intraperitoneal injections of compounds of comparable properties as ZM241385 to outline why we believe that ZM241385 in the way used in our study is likely to preferentially engage A2aR.

Results section:

‘Comparison with other emulsified compounds of a similar molecular weight suggests that such treatment led to a peak blood concentration of ZM 241385 in the low μ M range with a half-life of about 1h (34). At that peak concentration ZM 241385 inhibits both A2aR and A2bR (35). However, A2aR mRNA expression in T cells is several-fold higher than that of A2bR, A2aR-deficient lymphocytes don’t elevate cAMP in response to adenosine any more (20) and ZM 241385 displays more than 50-fold selectivity for A2aR over A2bR as detailed in the methods section. For the remainder of the manuscript we, therefore, refer to ZM 241385 as an A2aR antagonist.’

Methods section:

‘ZM 241385 properties, including selectivity for adenosine receptors, are detailed at <https://www.abcam.com/zm-241385-a2a-antagonist-ab120218.html>.’

We now also consistently state the ZM241385 concentration of 1.25 μ M used *in vitro* – apologies for not having done so earlier. We have extensively considered using SCH442416 up to running a first *in vivo* pilot experiment several years ago. But in discussion with members of the Cronstein laboratory we reached the conclusion at that time that ZM241385 may be more straightforward to administer *in vivo* than SCH442416. While we are aware that SCH442416 has now been used successfully *in vivo*, we believe that proper acknowledgement of receptor cross-reactivity of ZM241385 as detailed above may obviate the need to corroborate our findings with SCH442416.

In some experiments, the A2A agonist CGS21680 is used, while the nonselective agonist NECA is used in some other experiments. Again, it should be noted that NECA and ZM241385 are A2B agonist and antagonist, respectively. Although the potential role of A2B in myeloid cells is mentioned (ref. 38), it seems that role of A2B may not be limited to those cells, which could be discussed in this manuscript.

We now discuss this cross-reactivity in the corresponding results section:

‘This enhancement was reversed by parallel treatment with ZM 241385 at 1.25 μ M. NECA and ZM 241385 at the concentrations used engage both A2aR and A2bR (35). However, as A2aR mRNA expression in T cells is several-fold higher than that of A2bR and A2aR-deficient lymphocytes don’t elevate cAMP in response to adenosine any more (20), the induction of CL4 T cell translocation and the reversion thereof are most likely mediated by A2aR.’

Fig. 6, the concentration of CGS21680 used is 1 μ M, is it the same concentration used in other experiments?

We now consistently list the concentration of CGS21680 used, 1 μ M - apologies for not having done so earlier.

The authors could cite the ref. "J Biol Chem. 2003 Sep 26;278(39):37545-52. doi: 10.1074/jbc.M302809200. Epub 2003 Jul 1."

We now cite this paper in the discussion.

Reviewer 3

1. Line 72-81: Consider citing a recent article by Stone et al., demonstrating Treg depletion by non-blocking CTLA-4 mAbs as a dominant mechanism of CTLA-4 checkpoint blockade, <https://www.biorxiv.org/content/10.1101/2021.07.12.452090v1>

We now discuss this paper in the introduction.

2. Line 67: Related to above, when referring to mAb immunotherapy, it is imperative that the difference in the isotype of mAb and their interactions with Fc γ receptors (Fc γ R) are taken into account, as they may induce very different effects depending on the context, eg. <https://www.sciencedirect.com/science/article/pii/S1535610815002950>

We now mention the importance of Fc receptor binding in the introduction and, in response to your suggestion 10, in the first paragraph of the discussion as well (see above).

3. Fig 2: In addition to PCA data presented in Fig 2E, Fig 2D should show representative raw FACS plots of co-inhibitory marker expression data for both ZM 241385-treated and ctrl mice.

We now do so.

4. Majority of the numbers are not visible in Fig 2E.

We have changed the color of the numbers to red to increase visibility.

5. Line 268 & 357: correct 'didn't' to 'did not'. Same for 'doesn't' (line 575).

This is now corrected.

6. Discussion section: Given the potential translational nature of this study, it would be helpful if the authors could comment on the similarities/differences between murine and human receptors described herein. Importantly, would engagement/blockade of A2aR and TIM3 induce similar effects in a human culture setting?

We now address this question in the first paragraph of the discussion (see above).

7. Related to above, it would be important to comment on the current status of anti-TIM3 (eg, <https://www.nature.com/articles/s41577-019-0224-6>) and anti-A2aR clinical trials for human cancers and how this study may inform the design and development of future

therapeutics targeting such pathways.

We now address this question at the end of the second paragraph of the discussion.

8. FACS plots should be better formatted (eg, deletion of axis and gate labels) and the axis may adjusted for a better representation of the indicated cell populations.

We have deleted the axis labels and only retain the percentage of positive cells in the chosen gate.

9. General comment for Figures: Control group should be labelled and plotted first and more prominent and distinct colours and/or distinct lines (eg, solid vs. dotted or thicker lines) for the respective groups should be used throughout, eg, in Fig 5B, the colours for TILs, CTLs and the combination group are similar and the solid lines are narrow and in Fig 5B the symbol colours are very similar.

We thank the reviewer for these suggestions. We have now reordered group labels across all figures to list control groups first, and we have introduced a dotted line in Fig. 5 to allow more straightforward distinction between experimental groups. Our color scheme is consistent across the manuscript with the key *in vivo/in vitro* controls in black/grey, respectively. This should be sufficiently distinct to allow clear identification of the control groups.

10. In an ideal situation, deglycosylated TIM3 and PD-1 mAbs that lack any binding to FcγRs should be used for blocking experiments to prevent additional levels of complexity when interpreting the data (see above and the following examples: <https://www.sciencedirect.com/science/article/pii/S1535610816303920>). Furthermore, the ZM 241385 treatment group should have ideally been treated with an irrelevant isotype control mAb (ratIgG2a) for completeness, when compared to ‘ZM 241385 + anti-Tim3’ treatment group.

We thank the reviewer for raising these important caveats. We now discuss potential Fc receptor binding in the first paragraph of the discussion section (see above). We have also revised the methods section to highlight that we have not used an isotype control in the ZM241385 group, and we refer to published data where we have not seen any differences in Renca tumor growth *in vivo* between mice treated with buffer only versus isotype control: ‘Thus, ZM 241385-treated mice did not receive the mouse isotype control antibody. However, earlier work has shown that mouse isotype control antibody does not affect Renca tumor growth *in vivo* (7).’

REVIEWERS' COMMENTS:

Reviewer #1 (Remarks to the Author):

The authors have done a reasonably good job responding to my previous critiques. I have no further critiques.

Reviewer #2 (Remarks to the Author):

I do not have further comments.